# Targeting Mitochondrial Dysfunction in Cerebral Ischemia: Advances in Pharmacological Interventions

**DOI:** 10.3390/antiox14010108

**Published:** 2025-01-18

**Authors:** Igor Belenichev, Olena Popazova, Nina Bukhtiyarova, Victor Ryzhenko, Sergii Pavlov, Elina Suprun, Valentyn Oksenych, Oleksandr Kamyshnyi

**Affiliations:** 1Department of Pharmacology and Medical Formulation with Course of Normal Physiology, Zaporizhzhia State Medical and Pharmaceutical University, 69000 Zaporizhzhia, Ukraine; i.belenichev1914@gmail.com; 2Department of Histology, Cytology and Embryology, Zaporizhzhia State Medical and Pharmaceutical University, 69000 Zaporizhzhia, Ukraine; 3Department of Clinical Laboratory Diagnostics, Zaporizhzhia State Medical and Pharmaceutical University, 69000 Zaporizhzhia, Ukraine; 4Department of Medical and Pharmaceutical Informatics and Advanced Technologies, Zaporizhzhia State Medical University, 69000 Zaporizhzhia, Ukraine; 5The State Institute of Neurology, Psychiatry and Narcology of the National Academy of Medical Sciences of Ukraine, 46 Academician Pavlov Street, 61076 Kharkov, Ukraine; 6Faculty of Medicine, University of Bergen, 5020 Bergen, Norway; 7Department of Microbiology, Virology and Immunology, I. Horbachevsky Ternopil State Medical University, 46001 Ternopil, Ukraine; kamyshnyi_om@tdmu.edu.ua

**Keywords:** mitochondrial dysfunction, ROS, cerebral ischemia, HIF-1, HSP70

## Abstract

The study of mitochondrial dysfunction has become increasingly pivotal in elucidating the pathophysiology of various cerebral pathologies, particularly neurodegenerative disorders. Mitochondria are essential for cellular energy metabolism, regulation of reactive oxygen species (ROS), calcium homeostasis, and the execution of apoptotic processes. Disruptions in mitochondrial function, driven by factors such as oxidative stress, excitotoxicity, and altered ion balance, lead to neuronal death and contribute to cognitive impairments in several brain diseases. Mitochondrial dysfunction can arise from genetic mutations, ischemic events, hypoxia, and other environmental factors. This article highlights the critical role of mitochondrial dysfunction in the progression of neurodegenerative diseases and discusses the need for targeted therapeutic strategies to attenuate cellular damage, restore mitochondrial function, and enhance neuroprotection.

## 1. Introduction

The relevance of the study of various types of cerebral pathology and the development of methods for their treatment does not require detailed justification [1]. The World Health Organization defines human death as the death of the brain, which in life controls all the most important functions. In terms of prevalence and mortality, brain diseases rank third among diseases of the population of industrialized countries, leading not only to a decrease in life expectancy of the population but also limiting the social activity of a person due to the development of cognitive deficit, a decrease in the individual’s ability to think, learn, adequately perceive information, and make decisions [2,3]. In brain diseases of destructive and degenerative genesis, mitochondrial respiratory chain, energy metabolism, ionic homeostasis of the cell with increased content of calcium ions, development of glutamate excitotoxicity and damaging effect of nitrosative and oxidative stress, initiation of neuroapoptosis, and cell death occur [4]. The primary source of reactive oxygen species (ROS) is mitochondria, which play a key role in the cell’s energy supply [5]. Nowadays, there is a generalized concept of “mitochondrial dysfunction”. This is a typical pathological process that has no etiologic and nosologic specificity. The development of mitochondrial dysfunction eventually leads to neuron death [6]. We can speak about mitochondrial dysfunction as a new pathobiochemical mechanism of neurodegenerative disorders of a wide spectrum. All of the above is the rationale for the search for highly effective cerebroprotective drugs capable of preventing negative processes of mitochondrial dysfunction in the cell, thereby having a cerebroprotective effect [7]. Currently, energy-tropic drugs such as coenzyme Q10, carnitine, B vitamins, succinic acid derivatives, etc., are being tried to correct mitochondrial dysfunction [6,8]. However, the rational basis for their use is poorly developed, and effective approaches are often underutilized or ineffective ones are overestimated. Medications are applied chaotically, without sufficient knowledge of their potential and characteristics, and without planning a treatment strategy from the perspective of appropriateness. Moreover, in cases of established mitochondrial dysfunction and the activation of apoptotic processes, these drugs are largely ineffective as they cannot regulate the subtle mechanisms of energy metabolism in which they act as intermediates.

Another approach to correcting mitochondrial dysfunction is also being considered: the use of thiol antioxidants. These compete with the SH-groups of the cysteine-dependent site of the mitochondrial inner membrane protein (ATP/ADP antiporter) for ROS and peroxynitrite, forming stable complexes with the latter. This prevents the opening of the mitochondrial pore under conditions of oxidative and nitrosative stress.

Particularly interesting and deserving of special attention is the use of drugs that act as ligands for neuropeptide receptors. These can regulate apoptosis, the expression of transcription factors, the synthesis of enzymes that regenerate mitochondrial DNA, and enzymes that catalyze energy reactions. Recently, significant efforts have been directed toward identifying highly effective neuroprotectors among neuropeptides.

The above highlights the scientific appeal of studying mitochondrial dysfunction in neurons, making the development of new approaches to pharmacological correction highly relevant and promising from the standpoint of comprehensive neuroprotection [6,9,10].

### 1.1. The Role of Mitochondria in the Energy Metabolism of the Brain

The relationship between the structure and function of mitochondria remains a central focus of attention for a wide range of researchers. In recent years, revolutionary developments have taken place and are continuing in mitochondrial biology. Traditional views on the role of mitochondria in the cell have undergone significant revisions. Until relatively recently, our understanding of mitochondria was limited to their function as the cell’s energy stations, and the role of mitochondria in the development of pathology was confined to energy supply disruption, with all ultrastructural changes in mitochondria being viewed exclusively from this perspective. However, many ultrastructural states of mitochondria cannot be explained within such frameworks. Today, there is no doubt about the crucial significance of mitochondria for the life of eukaryotic cells. The dominant role of these organelles in ATP production, the execution of programmed cell death processes, their involvement in the generation of ROS, and the storage of calcium ions all determine their participation in the development of many pathological processes [6,11].

The brain has extremely high metabolic needs. It uses approximately 20% of the body’s total oxygen and glucose consumption with only 2% of its body weight. About 70% of estimated energy expenditure is used to support neuronal signaling, including resting potentials, action potentials, postsynaptic receptor activation, glutamate cycling, and postsynaptic Ca^2+^ signaling, while the remainder goes to nonsignaling activities such as biomacromolecule trafficking, axonal transport, mitochondrial proton leakage, and actin cytoskeleton remodeling [12,13,14].

Neurons exhibit the majority of energy consumption. They generate ATP primarily in mitochondria via oxidative phosphorylation, with a small portion of ATP from aerobic glycolysis in the cytoplasm. Astrocytes are highly glycolytic and convert glucose to lactate with low oxygen consumption; lactate is then delivered to neurons for complete oxidation. This process largely supports the energy requirements of neurons by supplying metabolic substrates [15,16].

Oligodendrocytes also obtain ATP mainly through aerobic glycolysis. They use lactate for their own energy needs and also supply neighboring axons with lactate. Microglia are mainly nourished by oxidative phosphorylation but are metabolically reprogrammed to a phenotype dominated by aerobic glycolysis under certain neurological circumstances [17,18,19]. Astrocytes have low mitochondrial activity of oxidative phosphorylation processes; this metabolic mode of astrocytes is essential for brain lipid homeostasis. Aberrant astrocytic oxidative phosphorylation process can cause accumulation of lipid droplets with subsequent development of neurodegeneration [20].

### 1.2. Concept of Mitochondrial Dysfunction

To date, on the basis of experimental studies, the concept of mitochondrial dysfunction, its formation, molecular, biochemical and ultrastructural features has been formulated [6,21,22,23].

The development of mitochondrial dysfunction leads to the disruption of neurotransmitter reuptake (catecholamines, dopamine, serotonin), ion transport, impulse generation and conduction, de novo protein synthesis, translation and transcription processes; “parasitic” energy-producing reactions are activated, resulting in a significant depletion of the neuronal cell’s energy reserves. Additionally, under the influence of ROS, particularly the hydroxyl radical, permeability transition pores are opened, leading to the expression and release of pro-apoptotic proteins into the cytosol. The opening of these pores occurs due to the oxidation of thiol groups in the cysteine-dependent region of the inner mitochondrial membrane protein (ATP/ADP antiporter), transforming it into a permeable nonspecific channel-pore. Today, the primary mechanism for this process is recognized as the formation of mitochondrial apoptotic pores and pores of increased permeability (PTP—permeability transition pores) [6,24,25,26,27].

The opening of the permeability pore typically allows the release of molecules up to 1500 Da. This disrupts mitochondrial metabolism, halting the synthesis of mitochondrial proteins and the import of proteins synthesized in the cytosol. Oxidative phosphorylation becomes uncoupled, and ATP synthesis stops. Hyperproduction of O_2_^−^ begins, and reducing equivalents are depleted. The opening of the pore transforms the mitochondria from “power plants” into “furnaces” for oxidation substrates without ATP production [28,29].

It was found that interaction of mitochondrial structures with active derivatives of NO and ROS, Ca^2+^ “overbreathing”, reduction of intramitochondrial GSH enhances pore opening and release of apoptogenic proteins from damaged mitochondria [30,31]. In this context, the role of one of the neurotrophic factors, tumor necrosis factor-α (TNF-α), with which the opening of pores in mitochondria, the subsequent disruption of their membranes and the development of mitoptosis are associated, is significant [32].

Thus, we can speak about mitochondrial dysfunction as a new pathobiochemical mechanism of neurodegenerative disorders of a wide spectrum. At present, two types of mitochondrial dysfunction are distinguished—primary as a consequence of congenital genetic defect and secondary, arising under the influence of various factors: hypoxia, ischemia, oxidative and nitrosative stress, expression of proinflammatory cytokines. In modern medicine, the doctrine of polysystemic disorders of cellular energy exchange, the so-called mitochondrial pathology, or mitochondrial dysfunction, occupies an increasingly important position [6].

A key area of this section of medicine is hereditary syndromes based on mutations of genes responsible for mitochondrial proteins (Kearns-Sayre, MELAS, MERRF, Pearson, Barth, etc.) [33].

However, the class of conditions characterized by mitochondrial dysfunction is by no means limited to these primary mitochondrial dysfunctions. A huge number of diseases include disorders of cellular energy metabolism—secondary mitochondrial dysfunctions as important links in pathogenesis. Among them: intracerebral hemorrhage, epileptogenic seizures, localized thermal brain damage, neurodegenerative disorders, transient cerebral ischemia, chronic fatigue syndrome, migraines, cardiomyopathies, alcoholic encephalopathies, senile dementia, neuroinfections, cardiomyopathies, glycogenoses, connective tissue diseases, diabetes, rickets, tubulopathies, pancytopenia, hypoparathyroidism, liver failure, and many others (Figure 1) [6].

The study of these disorders is of particular importance for practical medicine due to the unavailable effective therapeutic correction methods. However, it should be noted that the range of pathological disruptions in cellular energy metabolism is extremely wide (damages to various links in the Krebs cycle, respiratory chain, beta-oxidation, etc.). Mitochondrial dysfunction is closely associated with the hyperexpression of early genes, such as c-fos. In conditions of ROS hyperproduction by the neurochemical and bioenergetic systems of the brain during brain ischemia, as well as in several other neurodestructive pathological processes, there is an activation of the expression of redox-sensitive genes, many of which are essential for protecting cells from the toxic effects of oxidative stress [34,35,36].

Thus, under normal oxygen concentration in the surrounding cell environment (normoxia), the activation of JunB and ATF-2 transcription factors mainly occurs under the action of ROS, while under oxidative stress, the activation of c-Jun and c-fos factors predominates. The activation of these specific transcription factors under conditions of ROS hyperproduction is explained by the fact that JunB and c-fos contain cysteine residues (Cys252, Cys54, Cys61) in their DNA-binding domains that are highly sensitive to ROS. Oxidation of their SH-groups leads to the reverse inactivation of AR-1 and NF-kB [6,37,38].

In addition, c-fos protein is directly involved in the process of mitochondrial DNA fragmentation and initiation of apoptotic neuronal cell death. c-fos is responsible for NO hyperproduction in neurodegenerative diseases through iNOS activation. c-fos is one of the main nuclear targets for signaling regulation of cell growth and transformation and is involved in many cellular functions, including cell proliferation and differentiation processes [39,40,41].

It is currently known that the main manifestations of mitochondrial dysfunction are a decrease in ATP levels in the cell, activation of cell death mechanisms, and the production of ROS by the mitochondria. It is known that during the functioning of the mitochondrial respiratory chain, small amounts of superoxide radical (O^•−2^) are formed as a byproduct of the respiratory complexes [42,43].

## 2. Molecular Mechanisms Behind Mitochondrial Dysfunction

### 2.1. Markers of Mitochondrial Dysfunction

One important mitochondrial control system that has recently been recognized is the sirtuins. These conserved class III histone deacetylases possess a catalytic nicotine-adenine dinucleotide + (NAD+) binding domain and play an important role in the regulation of both inflammation and metabolism [44,45].

In mitochondria, sirtuins utilize NAD+ as a sensor of mitochondrial energy status, with SIRT3 enhancing oxidative stress control by increasing SOD levels, SIRT 4 enhancing the ATP/ADP transporter, and SIRT 5 enhancing urea cycle function [46,47].

As a marker of mitochondrial dysfunction we use AMA-M2—antimitochondrial antibodies directed to different proteins of the inner membrane of mitochondria. M2 is directed to the pyruvate dehydrogenase complex, and M3 to the malate dehydrogenase complex [48,49,50,51,52].

Also used as a marker of mitochondrial dysfunction, is Cytochrome C, which is synthesized as apocytochrome C and enters the mitochondrion where it binds to the inner surface of the membrane and then exits into the cytoplasm through channels that are opened for it by Bcl-2 family proteins [53,54,55].

Biochemical markers of mitochondrial dysfunction such as lactate and pyruvate are widely recognized. Pyruvate is a product of glycolysis, the pathway that metabolizes glucose for fuel, and is the first step in mitochondrial metabolism of carbohydrates. Pyruvate enters the tricarboxylic acid cycle, one of the main mitochondrial metabolic pathways. If there is mitochondrial dysfunction, the tricarboxylic acid cycle slows down, leading to pyruvate accumulation. As pyruvate accumulates, it is diverted to other pathways for metabolism, specifically lactate and alanine. Thus, when mitochondria become dysfunctional, three metabolic biomarkers can rise in the blood: pyruvate, lactate, and alanine. In fact, lactate and pyruvate are used in the diagnostic criteria for mitochondrial disease. The two ratios are also used to monitor mitochondrial function. The lactate to pyruvate ratio is elevated when lactate is overproduced from pyruvate. Another lesser-known ratio is the alanine-to-lysine ratio. Lysine is produced as a product of acetyl-CoA, the first step of the tricarboxylic acid cycle. Thus, when the tricarboxylic acid cycle slows down, less lysine and more alanine is produced, raising the alanine-to-lysine ratio to over 2.5 [56,57,58,59].

Carnitine is used as a biochemical marker of mitochondrial dysfunction. Low levels of total or free carnitine are markers of several diseases associated with primary mitochondrial dysfunction. Carnitine plays an important role in fatty acid metabolism but also serves another purpose. Carnitine can bind to fatty acids in the blood as acylcarnitines and can also bind to excess organic acids. This increases the amount of carnitine esters and decreases free carnitine. Carnitine can then be excreted in the urine along with these fatty acids or organic acids if in excess, resulting in a loss of carnitine in the body, reducing the total amount of carnitine. Short and medium-chain fatty acids can directly enter the mitochondria, while long-chain fatty acids are bound to carnitine and transported through the carnitine shuttle. Fatty acids are metabolized by β-oxidation, which is a cycle that shortens a fatty acid by two carbons with each turn, producing acetyl-CoA, which directly enters the first step of the tricarboxylic acid cycle, and FADH 2, which directly contributes to the electron transport chain as a substrate for complex II. Elevated levels of Acyl-CoA are markers of diseases associated with primary mitochondrial dysfunction (Figure 2) [60,61,62,63,64].

### 2.2. Mitochondrial Antioxidant System

In functionally complete mitochondria, the action of the mitochondrial antioxidant system, including glutathione, thioredoxin-2, glutathione peroxidase, phospholipid-hydroperoxide-glutathione peroxidase, and Mn-superoxide dismutase, prevents damage to mitochondrial structures by reactive oxygen species [65,66].

GSH is the most important antioxidant in the cell, and the ratio of GSH to GSSH is the most important biomarker of the redox state of the cell. GSH cannot be produced within mitochondria, so it must be imported. A pathway that produces more GSH requires ATP, an energy molecule produced by the mitochondria, so poor mitochondrial function will result in decreased GSH production and reduced ability of the mitochondria to produce cellular energy. ROS is controlled at the inner mitochondrial membrane by leakage of protons back across the membrane, a process that makes mitochondria less efficient. This is accomplished by several proton channels such as uncoupling protein (UCP) [67,68,69,70].

We found that the introduction of chloro-2,4-dinitrobenzene (CDNB, 1 mM) into the suspension of mitochondria isolated from the brain of white rats leads to a 62% decrease in the amount of GSH and more than 47% inhibition of Mn-SOD activity. A decrease in mitochondrial membrane potential and an increase in mitochondrial pore opening rate were also detected. Deprivation of the mitochondrial glutathione system leads to a decrease in other antioxidant systems as well and shapes the development of mitochondrial dysfunction. GSH deficiency in mitochondria leads to increased formation of ROS and nitrogen and oxidation of cysteine-dependent sites of proteins forming the mitochondrial pore. Excess of active nitrogen forms (peroxynitrite, nitrosonium ion) formed at GSH deficiency in mitochondria leads to oxidative modification of Mn-SOD and reduction of its activity. Reduction of Mn-SOD activity promotes a secondary “burst” of free-radical reactions and intensification of oxidative destruction of Red-Oxi—sensitive sites of mitochondrial membrane and formation of persistent mitochondrial dysfunction [71].

### 2.3. The Production of ROS by Mitochondria and the Lack of Mitochondria

It is believed that a key event in the development of mitochondrial dysfunction after hypoxia/reoxygenation is an increase in the production of ROS by mitochondria. The main causes of mitochondrial dysfunction include electron overload in the respiratory chain under hypoxic conditions, as well as a decrease in the activity of cytochrome c oxidase and Mn-superoxide dismutase (Mn-SOD). Inhibition of cytochrome c oxidase during subsequent reperfusion contributes to a disruption in the process of electron transfer to the final acceptor in the respiratory chain—oxygen—which leads to an increase in the production of superoxide radicals by the respiratory complexes [72].

Superoxide is formed in the so-called “parasitic” reactions in the initial part of the mitochondrial respiratory chain (CoQH2-NAD+) with the participation of NADH-CoQH2-reductase, the activity of which is increased by blockade of cytochrome-C-dependent receptor on the outer surface of the mitochondrial membrane against the background of increased reduced flavins. In addition to superoxide, the key role in the development of mitochondrial disorders and the development of apoptosis/necrosis belongs to NO and its more aggressive form—peroxynitrite [73,74].

Neuronal mitochondria are an important source of NO. The presence of a constitutive form of NOS localized in the inner membrane and NO production in mitochondria of hippocampal neurons has been shown. Mitochondrial NOS at suboptimal concentrations of L-arginine is able to produce superoxide. Mitochondrial NOS (mtNOS) is significantly activated in response to the development of glutamate “excitotoxicity” and calcium uptake by mitochondria [75,76].

It should be noted that IL-1b and TNF-a play a certain role in mtNOS activation. mtNOS through the production of a dosed level of NO, is able to regulate mitochondrial respiration in normal conditions and at the initial, compensated stages of ischemia, modulating the activity of cytochrome-C-oxidase, complexes I and II of the electron-transport chain and the level of NADPH, FAD, and coenzyme Q10, as well as changing the availability of O_2_ for electron acceptance. Further, the role of mtNOS changes to the cardinally opposite one—it participates in activation of “parasitic” reactions of ROS formation by mitochondria. The thiol-disulfide system deserves special attention in expanding the understanding of the mechanisms of NO cytotoxicity and neuronal death [77,78,79].

Intermediates of the thiol-disulfide system have transport properties with respect to NO, thereby increasing its bioavailability; moreover, many thiols such as glutathione, cysteine, and methionine can significantly limit the cytotoxicity of NO and its derivatives, increasing the chance of neuronal mitochondria to survive ischemia [26,80,81].

### 2.4. Interaction of ROS and NO with the Mitochondria

When electron transfer between the components of the respiratory chain is disrupted, the generation of superoxide anion (O^•−2^) by mitochondria is significantly enhanced. A deficiency in the functions of the mitochondrial antioxidant system will contribute to the development of oxidative stress, activation of self-sustaining processes of lipid peroxidation, and oxidative damage to proteins and nucleic acids within the mitochondria [82].

The increase in NO concentration in mitochondria observed in the post-ischemic period leads to the interaction of NO with heme iron and paired thiol groups, forming dinitrosol iron complex (DNIC). DNIC, in contrast to NO, is a stronger nitrosylating agent, interacting with protein thiols, histidine, aspartate, glutamine, methionine, cysteine, glutathione and forming N- and S-nitrosothiols [83].

Cerebral ischemia is accompanied by a sharp shift of thiol-disulfide equilibrium towards oxidized thiols, a drop in the activity of enzymes of the thiol-disulfide system (glutathione reductase, glutathione-S-transferase). When glutathione-dependent enzymes are inhibited under ischemia conditions, even greater oxidative modification of low molecular weight thiols occurs, homocysteine formation and, as a consequence, NO transport is impaired with the formation of its cytotoxic derivatives that further enhance thiol oxidation. The presence in a neuron of a sufficiently active thiol antioxidant system capable of regulating NO transport provides cell resistance to nitrosative stress, the earliest neurodegenerative mechanism under ischemia conditions [84,85,86]. It is known that in the first minutes of brain ischemia, NO (macrophagal or exogenous) inhibits oxidative phosphorylation in mitochondria of target cells due to reversible binding to mitochondrial cytochrome C-oxidase. Suppression of electron transport in mitochondria, as shown above, leads to generation of superoxide and, as a consequence, to the formation of ONOO- [26].

Excess superoxide radical, peroxynitrite further oxidizes the thiol groups of the cysteine-dependent site of the mitochondrial inner membrane protein (ATP/ADP-antiporter), which turns it into a permeable nonspecific channel pore. The mitochondrial pore is a supramolecular channel connecting the cytosolic and intramitochondrial spaces, composed of a complex of proteins including the adenine nucleotide translocator, the benzodiazepine receptor (translocator protein), and the voltage-dependent anion channel. Cyclophilin D, ATPase, and the mitochondrial inorganic phosphate carrier are not part of the pore structure but act as regulatory factors. The anti-apoptotic bcl-2 and pro-apoptotic Bax proteins are associated with the benzodiazepine receptor. These proteins are components of the outer membrane and cytosol and participate in apoptosis by controlling the release of cytochrome C [87,88].

Under conditions of oxidative stress, the mitochondrial pore binds to Ca^2+^ and substances with large molecular weight can pass through the membrane pore. This leads to a drop in membrane potential and matrix swelling, the integrity of the outer membrane is inevitably compromised, and apoptosis proteins are released from the intermembrane space into the cytoplasm [89].

There are several pro-apoptotic proteins: apoptosis-inducing factor (AIF), second mitochondria-derived activator of caspases (Smac), and some procaspases. The inducing factor goes directly to the nucleus, where it causes DNA degradation. Along with specific apoptosis proteins, cytochrome C, which normally serves as the final link in the electrotransport chain, leaves the mitochondrion through the open pore. In the cytoplasm, this protein binds to the protein Apaf-1 (apoptotic protease activating factor-1) and forms an apoptosome complex. With the help of Smac and another factor (Omi/HtrA2), it activates procaspase-9, which, becoming caspase-9, transforms two other proenzymes into caspases-3 and -7; and they already cleave structural proteins, leading to the appearance of biochemical and morphological signs of apoptosis in the neuronal cell [90,91].

### 2.5. Disorder in the Lipid Layer of Mitochondrial Membranes

In parallel with the above-mentioned destructive processes in the mitochondrial matrix, oxidative changes in the lipid layer of mitochondrial membranes are observed: the level of phospholipids decreases and free fatty acids and lysophosphatides accumulate. Depletion and oxidative modification of mitochondrial pool of membrane phospholipids promote mobilization of cytochrome C in the intermembrane space, which significantly facilitates its release into the cytoplasm after opening of mitochondrial pores [92,93].

It was found that changes in the lipid composition of mitochondrial membranes lead to impaired functioning of mitochondrial membrane enzymes. The increase in the content of free fatty acids in the inner mitochondrial membrane promotes dissociation of oxidation and phosphorylation processes, which leads to suppression of ATP synthesis [94,95].

ROS are produced through metabolic processes, particularly by Complexes I and III of the electron transport chain (ETC). ROS can be destructive to many vulnerable parts of the mitochondria. Lipid membranes are especially vulnerable to ROS. The integrity of lipid membranes is crucial for key mitochondrial structures, such as the ETC. If the lipid membrane is damaged, protons may leak through the inner mitochondrial membrane, which can reduce the proton gradient responsible for driving Complex V of the ETC to produce ATP. This will decrease ATP production and lower the inner mitochondrial membrane potential [42,96].

## 3. Genetic and Cellular Factors Involved in Mitochondrial Dysfunction

### 3.1. Mitochondrial DNA Damage in Mitochondrial Dysfunction

Another mechanism of mitochondrial dysfunction is the accumulation of damage in the mitochondrial genome and the depletion of the mitochondrial DNA pool. ROS can also damage mitochondrial DNA (mtDNA), as mtDNA is not protected like nuclear DNA, and mitochondria are inefficient in repairing mtDNA. This can lead to harmful mutations in mtDNA if the damage occurs in a critical genetic region of the mtDNA. Ultimately, these events lead to a decrease in cell function and the accumulation of mutations in both mitochondrial and nuclear DNA. In turn, damage to mitochondrial genes contributes to the disruption of the electron transfer process in the respiratory chain, resulting in further increased production of free radicals in the mitochondria [97,98].

ROS, interacting with nucleic acids, modify bases, deoxyriboses, and also form new covalent bonds. The most significant modification occurs at the bases [99]. When ROS and free radicals interact with thymine, 5,6-dihydroxy-5,6-dihydrothymine isomers are formed. Cytosine, when interacting with ROS and hydroperoxides, forms hydroxylated cytosine. The interaction of ROS with purines leads to the breakage of the imidazole ring of the molecule fragment, resulting in the formation of formamidopyrimidine residues [100].

Among the products of oxidative modification of purines, the most notable are 8-oxoguanine (8-OG) and its tautomer 8-hydroxyguanine (8-OHG). The formation of these products occurs continuously, but it significantly increases under various pathological conditions and can be considered a marker of oxidative stress. Currently, the determination of 8-OHG is a popular, non-invasive method for assessing oxidative stress levels. Under the influence of ROS, deamination of guanine and adenine may occur, leading to the formation of xanthine and hypoxanthine in the DNA molecule, which have mutagenic effects. The consequences of various types of oxidative damage are not the same. For example, thymine glycols and formamidopyrimidines block replication, are cytotoxic, but their mutagenic potential is limited. The most mutagenic is 8-OHG, and if it is present in the template, all replicative DNA polymerases insert dAMP opposite it and carry out a guanine-cytosine to thymine-adenine substitution. In addition, ROS can directly induce DNA strand breaks by cleaving the sugar-phosphate base [101,102]. DNA aberration induces an enhanced stress response in the mitochondria, which ultimately leads to the formation of defective mitochondrial proteins. ROS are capable of causing indirect damage to nucleic acids. ROS trigger the release of calcium from the mitochondria, which subsequently leads to an increase in nuclease activity. Nitric oxide (NO) plays a significant role in DNA damage by causing the deamination of nucleic acids, followed by the activation of NAD-dependent polymerase (PARP), which catalyzes the attachment of ADP-ribose to histone proteins and DNA [103,104].

Additionally, it should be noted that transporting 1 mole of ADP-ribose requires 1 mole of NAD+ and 4 moles of ATP, which, with significant mobilization of PARP following extensive DNA damage, rapidly depletes the cell’s energy reserves. ROS also stimulate ADP-dependent ribosylation of glyceraldehyde-3-phosphate dehydrogenase, leading to the subsequent inactivation of the enzyme and disruption of glycolysis reactions. The impairment of the mitochondria’s energy-producing function is also associated with the dysfunction of the respiratory chain, caused by mutations in mitochondrial DNA (mtDNA). Such disruptions affect various biochemical functions of the mitochondria, such as the mitochondrial membrane potential, ATP synthesis, the ATP/ADP ratio (which indicates the state of the oxidative phosphorylation system), ROS generation, and mitochondrial turnover. It has been established that once the proportion of ROS-mediated mtDNA deletions exceeds a sensitivity threshold, the mitochondrial membrane potential, ATP synthesis rate, and ATP/ADP ratio sharply decrease [105,106,107].

The main consequences of the aforementioned processes are the disruption of the respiratory chain function, weakening of the mitochondrial antioxidant defense, accumulation of cytotoxic oxidatively damaged proteins and nucleic acids; mobilization of cytochrome C in the intermembrane space, which, after reoxygenation, transforms the mitochondria into a source of free radicals, reduces their ability to synthesize ATP, and increases the mitochondria’s sensitivity to thanatogenic signals [108,109].

### 3.2. Ca^2+^ and Mitochondrial Dysfunction

A significant impact on the mitochondria is exerted by the increase in the cytoplasmic level of calcium ions due to the disruption of the cell’s ionic homeostasis. It has been shown that the increase in calcium ion concentration in the cytoplasm promotes the induction of the release of thanatogenic factors from the mitochondria, initiates lipolysis processes in the mitochondrial membranes, and disrupts the function of respiratory complexes. It has been established that Ca^2+^ ions activate proteins that facilitate the formation of mitochondrial channels: Bax and Bid, by activating calpains, as well as Bad and Bik, through the activation of calcineurin [110,111,112].

Movement of excessive amounts of calcium ions into the mitochondrial matrix of Ca^2+^ ions can lead to additional opening of permeability transition pores. Another of the known mechanisms of the effect of excessive concentration of Ca^2+^ ions on mitochondrial structures is damage to the mitochondrial membrane due to activation of phospholipase A2 [113]. The possibility of stimulation of oxidative stress in the cell by Ca^2+^ ions through activation of calcium-sensitive isoform of NOS has been shown [114]. It is known that uncontrolled increase in the concentration of neurotransmitters such as glutamate and dopamine in the extracellular space contributes to the development of neurotoxic processes in the affected area of the brain during stroke. It has been established that excessive glutamate induces unregulated Ca^2+^ ion influx into the cytosol, further impairing the functional activity of mitochondria [115]. Glutamate is the main excitatory neurotransmitter of the CNS, is involved in cognitive functions, along with acetylcholine maintains the level of wakefulness, but in high concentrations is a neurotoxin [116].

Glutamate realizes its effects through a group of ionotropic membrane receptor channels: NMDA, AMPA, and kainate receptors. Excitation of glutamate NMDA receptors, which regulate the content of K^+^, Na^+^, Ca^++^, Cl^−^ in the extra- and intracellular space, activates Ca^++^-channels, which leads to an increase in the flow of extracellular Ca^++^ into the cell and release of intracellular Ca^++^ from the depot, activating various enzyme systems [117]. This leads to impaired phosphorylation of proteins, cleavage of phospholipids and release of arachidonic acid, formation of toxic products, free radicals that have cytotoxic, immunogenic and mutagenic effects, damaging cellular DNA and mRNA [118].

Along with the swelling of mitochondria caused by the influx of calcium, the process shifts into the cytoplasm of the cell and extends to the intercellular level, making the hypoxia tissue-wide. This stage of the ischemic cascade can no longer be reversed by restoring oxygen supply or reperfusion, as the deeply damaged mitochondria stop utilizing oxygen and substrates. They combine with sodium and calcium in the cytoplasm to form endogenous soaps, which literally dissolve (wash away) lipid membranes [119,120,121,122].

### 3.3. Disruption of Native Protein Structure (Unfolded Protein Response—UPR) and Mitochondrial Dysfunction

An additional factor contributing to mitochondrial damage is the “unfolded protein response” (UPR), which is activated under hypoxic conditions. It has now been shown that the endoplasmic reticulum (ER), when under stress mediated by UPR, can promote the development of degenerative changes in mitochondria by releasing calcium ions into the cytoplasm from the ER, along with the membrane-associated ER protein inositol-requiring enzyme 1 (IRE1). The ER stress sensor IRE1 interacts with STIM1, facilitating the influx of Ca^++^ and initiating calcium-dependent responses, including the activation of NOS-regulated proteins [123,124,125].

UPR normally mediates cell death by activation of the intrinsic apoptotic pathway, but recent studies have shown that in mitochondrial dysfunction there is a strong activation of UPR, which may lead to activation of programmed necrosis pathways such as necroptosis [126]. To date, it is known that mitochondrial dysfunction is possible through inactivation of the hypoxia-inducible transcription factor HIF-1 [127].

Recent studies have established that adaptation to hypoxia at the cellular and subcellular levels is closely associated with the transcriptional expression of late-acting hypoxia-inducible genes, which are involved in regulating multiple cellular and systemic functions and are necessary for the formation of adaptive traits. It is known that at low oxygen concentrations, this process is primarily controlled by the specific transcription factor HIF-1, which is induced by hypoxia in all tissues. Disruption of energy metabolism due to ischemia or hypoxia, and a decrease in the concentration of mitochondrial ATP, leads to the activation of mitochondrial UPR (UPRmt). In the early stages of ischemia, UPRmt “attempts” to restore mitochondrial energy homeostasis, and if unsuccessful, it suppresses the expression of protective proteins, including HIF-1, and initiates cell death mechanisms [128,129,130]. The use of novel specific UPR inhibitors as remedies for mitochondrial dysfunction is of interest.

### 3.4. HIF-1 and Mitochondrial Dysfunction

Hypoxia-induced factor, discovered in the early 90s, functions as a master regulator of oxygen homeostasis and is the mechanism by which the organism, responding to tissue hypoxia, controls the expression of proteins responsible for the mechanism of oxygen delivery to the cell, i.e., regulates the adaptive responses of the cell to changes in tissue oxygenation [131,132,133].

Currently, more than 60 direct target genes have been identified for it. All of them contribute to the improvement of oxygen delivery (erythropoiesis, angiogenesis), metabolic adaptation (glucose transport, enhanced glycolytic ATP production, ion transport) and cell proliferation. HIF-1 regulated products act at different functional levels. The end result of such activation is an increase in O_2_ supply to the cell [119,134].

The identification and cloning of HIF-1 have revealed that it is a heterodimeric redox-sensitive protein composed of two subunits: the inducibly expressed oxygen-sensitive subunit HIF-1α and the constitutively expressed subunit HIF-1β (aryl hydrocarbon receptor nuclear translocator—ARNT). By heterodimerizing with the aryl hydrocarbon receptor (AHR), it forms a functional dioxin receptor. Other proteins of the HIF-1α family are also known, including HIF-2α and HIF-3α. All of them belong to a family of basic proteins, each containing a basic helix-loop-helix (bHLH) domain at the amino-terminal end of each subunit, which is characteristic of various transcription factors and essential for dimerization and DNA binding [135,136,137].

HIF-1α consists of 826 amino acid residues (120 kD) and contains two transcriptional domains at the C-terminal end. Under normoxic conditions, its synthesis occurs at a low rate and its content is minimal because it undergoes rapid ubiquitination and degradation by proteasomes. This process depends on the interaction of the primary structure of HIF-1α and its specific oxygen dependent degradation domain (ODDD—oxygen dependent domain degradation) with von Hippel Lindau (VHL), a tumor growth suppressor, which acts as a protein ligase, widely distributed in tissues [138,139,140].

The molecular basis for this regulation is the O_2_-dependent hydroxylation of its two proline residues P402 and P564, part of the HIF-1α structure, by one of three enzymes collectively known as “prolyl hydroxylase domain (PHD) proteins, or HIF-1α prolyl hydroxylases,” which is required for binding of HIF-1α to the VHL protein. α-Ketoglutarate, vitamin C and iron are also obligatory components of the process. Along with this, hydroxylation of an asparagine residue in the C-terminal transactivation domain (C-TAD) occurs, resulting in suppression of HIF-1α transcriptional activity. After hydroxylation of the proline residue in the ODDD and the asparagine residue, HIF-1α binds to the VHL protein, which makes this subunit available for proteasomal degradation [141,142].

Under conditions of severe oxygen deficiency, the oxygen-dependent process of hydroxylation of proline residues, which is characteristic of normoxia, is suppressed. As a result, VHL (von Hippel-Lindau protein) cannot bind to HIF-1α, and its degradation by the proteasome is limited, allowing for its accumulation. In contrast, p300 and CREB binding protein (CBP) can bind to HIF-1α, as this process does not depend on asparaginyl hydroxylation. This enables the activation of HIF-1α, its translocation to the nucleus, dimerization with HIF-1β, leading to conformational changes and the formation of a transcriptionally active complex (HRE), which triggers the activation of a broad range of HIF-1-dependent adaptive processes aimed at enhancing the synthesis of endogenous cytoprotective proteins. Among the most important proteins in this group are the so-called heat shock proteins, such as HSP70 (heat shock proteins) [143,144].

HIF-1 mediates mitochondrial biogenesis, mitophagy, and mitochondrial dynamics to regulate mitochondrial population. HIF-1 activation induced mitochondrial fission in human models of pulmonary arterial hypertension (PAH) by phosphorylation of DRP1 by serine 616 [145].

HIF-1 can alter the intracellular distribution of mitochondria by regulating their mobility. The mitochondrial movement regulator (HUMMR) is activated by HIF-1α, and mitochondrial transport shifts in the anterograde direction for efficient distribution throughout the neuron. Additionally, HUMMR may help preserve mitochondrial content in axons dependent on HIF-1α. HIF-1 regulates mitochondrial morphology, such as size, shape, and structure, which could underlie functional changes or be secondary to functional regulation. In mitochondria with stimulated HIF-1 expression, there was an increase in energy metabolism reactions and regulation of ROS production by mitochondria. HIF-1 positively regulates the expression of lactate dehydrogenase A (LDHA) and promotes the conversion of pyruvate to lactic acid under hypoxic conditions. HIF-1 serves as a sensor for ROS, limiting excessive mitochondrial ROS production in response to cytokine stimulation. Overexpression of HIF-1α blocks the reduction of mitochondrial membrane potential (ΔΨm) under hypoxia and inhibits mitochondrial mechanisms that initiate apoptosis [127,146,147].

There is evidence for a relationship between HIF-1α and iron homeostasis, especially at the mitochondrial level, because Fe^2+^ is mobilized in the Fenton reaction, which produces hydroxyl radical (^•^OH) from H_2_O_2_ and lipid alkoxyl radicals from lipid peroxides. Notably, the mitochondrial antioxidant apparatus allows iron homeostasis to be maintained by limiting H_2_O_2_ production and converting lipid peroxides to lipid alcohol using SOD and GPX4, respectively [127,148].

### 3.5. HSP70 and Mitochondrial Dysfunction

Enhanced expression of genes encoding HSP is triggered not only by heat stress but also by a number of different factors and pathogens. Evolutionarily, HSP70 is classified as a highly conserved protein, indicating that it performs fundamental cellular functions. The native HSP70 molecule is a dimer with the ability to form highly oligomeric complexes with many structures in the cell, as well as cytosolic and mitochondrial proteins, and has at least 8 isoforms, the exact number and concentration of which depends on the cell type and is controlled by the type of stressor. HSP70 proteins belong to a class of cellular proteins referred to as “molecular chaperones” (chaperone-mediator). Chaperone activity refers to the ability of HSP70 to recognize and bind exposed hydrophobic surfaces of native polypeptide chains, denatured and oxidatively damaged polypeptides. HSP70 are the main participants in the process of folding of newly synthesized polypeptide chains [149].

The ability of HSP70 to fold polypeptides in a specific conformation is used in normal cellular processes to regulate key signaling molecules such as cell cycle kinases, caspases, steroid hormone receptors, and vitamin D receptors. HSP70 are also involved in the processes of polypeptide translocation into mitochondria, in restoring the structure of damaged and denatured proteins, and in the formation of oligomeric protein complexes. In addition to chaperone function, HSP70 function as regulators/modulators of protein proteolysis (ubiquitins, redox reactions, synthesis of proinflammatory cytokines, synaptic transmission, and Ca^++^-dependent K^+^-channels). In addition, the ability of HSP70 to stabilize under ischemia conditions the factor HIF-1 was investigated. Thus, under normoxia conditions, HSP-70 is in complex with HIF-1 [150,151].

Under hypoxia conditions, HSP70 is displaced from the complex with HIF-1 by ARNT protein, with further exercise of its chaperone function against HIF-1, in addition, these proteins exert unidirectional action with respect to cell protection against oxidative stress during ischemia [149,152,153].

Recent studies have established the neuroprotective activity of HSP70 and HIF-1 aimed at reducing the phenomena of mitochondrial dysfunction and related oxidative stress. We found that HSP70 and HIF-1 in brain ischemia increase the activity of mitochondrial antioxidant enzymes protecting them from oxidative destruction, restore thiol-disulfide equilibrium, normalize the processes of energy metabolism due to folding of respiratory chain proteins, as well as increase the functional activity of mitochondria, eliminating damaged and denatured proteins. Start and regulate the activity of the compensatory malate-aspartate shuttle mechanism of energy production [153,154].

This statement is confirmed by the works of some authors. The role of increased expression of HSP70 in brain cells (astrocytes) in protecting them from death caused by oxygen starvation has been shown [155,156].

In addition, the ability of a purified HSP70 preparation to enhance the survival of neurons involved in glutamatergic synaptic transmission in the olfactory cortex of rat brain was demonstrated against the damaging effects of severe anoxia [157,158]. Nevertheless, the mechanism of the protective effect of HSP70 is still unclear. Taking into account the data on the ability of HSP70 to enhance neuronal cell viability under hypoxia conditions and the fact of interaction between HSP70 and HIF-1, which plays a primary role in the cellular response to hypoxia, we can assume that HSP70 is involved in the regulation of signaling pathways of the cell response to hypoxic stress at the level of regulation of HIF-1 stability [149].

In addition, the neuroprotective effect of HSP70 in ischemia is also explained by its antiapoptotic and “mitoprotective” action. Currently, three main pathways of HSP70 influence on apoptosis processes are postulated in the literature. Firstly, HSP70 may affect the functioning and signaling of the Fas/Apo1 receptor inside the cell; secondly, HSP70 may in one way or another affect the release of cytochrome C from mitochondria; and finally, thirdly, HSP70 may affect the formation of apoptosomes and the activation of the caspase cascade. HSP70 blocks apoptosis induced by activation of the Fas/Apo1 receptor. After binding to the ligand, the receptor interacts with adaptor proteins, one of which may be the FADD protein [159,160,161,162].

This adaptor protein FADD binds inactive procaspase 8 and promotes its activation upon receptor-ligand binding. Caspase 8 activates caspases 3, 6, and 7 and thereby initiates proteolysis of target proteins, ultimately leading to apoptosis. In addition, caspase 8 can activate the Bid protein, which induces the release of cytochrome c from mitochondria [163,164].

The site of action of HSP70 in this complex chain of reactions has not yet been precisely determined. An alternative pathway for triggering apoptosis via Fas/Apo1 involves the Daxx protein. The mechanism of action of this protein is not well understood. Normally, Daxx is localized in the nucleus, where it is bound to certain proteins, but it can move to the cytoplasm and play the role of an adaptor protein responsible for triggering the cascade of JNK-kinases by activating Fas/Apo [165,166].

It is assumed that HSP70 is able to move to the nucleus, where it interacts with Daxx, preventing its release into the cytoplasm and activation of the receptor. Previously, it was noted that HSP70 may participate in the regulation of apoptosis not only at the level of the receptor Fas/Apo1, but also at the level of certain intracellular target proteins [167,168].

Indeed, HSP70 has been shown to prevent mitochondria-initiated apoptosis, and different mechanisms of action of heat shock proteins are possible. The drop in membrane potential induced by cerebral ischemia is known to result in the release of cytochrome c from mitochondria. In the cytoplasm, cytochrome C binds to Apaf1 protein, deoxy ATP, and procaspase 9, forming the so-called apoptosome [169,170].

Apoptosome formation is accompanied by autocatalytic activation of procaspase 9 and its transfer into the active form of caspase 9. This enzyme activates procaspase 3 and the following caspases involved in the process of apoptosis. HSP70 inhibits apoptosis in the step between cytochrome c release and cleavage of procaspase 9 in the apoptosome. Recently, the literature has provided evidence that HSP70 is able to interact with cytochrome C [171,172]. The question of what part of cytochrome C released from mitochondria binds to HSP70 remains open. A number of studies have shown that HSP70 binds only a very small fraction of cytochrome C released from mitochondria and, therefore, cannot play a significant role in apoptosome formation [173,174]. HSP70 is known to prevent the decrease in membrane potential induced by Bax protein, but does not interact with this protein. There is a hypothesis that in the mitochondrial pathway of apoptosis HSP70 acts at earlier stages of this complex process and prevents disruption of the actin filament structure [154].

Our works established that modeling of chronic cerebral circulatory disturbance by disruption of cerebral blood circulation led to persistent neurological disorders in surviving animals by the 18th day of the experiment, as well as to the disruption of mitochondria ultrastructure of hippocampal CA1 neurons, which was characterized by an increase in the absolute number of damaged mitochondria, more than 11 times in relation to the intact group of animals, as well as a decrease in intramitochondrial HSP70 3,4 in comparison with similar indicators of the group of falsely operated animals [25].

Thus, summarizing the above, we can conclude that HSP70 and Hif1b proteins are inevitable companions of pathobiochemical reactions that develop during ischemic brain damage and perform a protective function under these conditions, which is realized by means of enhancing the synthesis of antioxidant enzymes, stabilization of oxidatively damaged macromolecules, direct antiapoptotic and mitoprotective action. Such a role of these proteins in cellular reactions during ischemia raises the question of the development of new neuroprotective agents capable of modulating the expression and synthesis of HSP- and HIF-proteins [6].

## 4. Mitochondrial Dysfunction and Apoptosis

The activation of neuroapoptosis, according to many researchers, is the primary cause of persistent cognitive and mnemonic dysfunctions in the central nervous system (CNS). Neuroapoptosis develops as a cascade process accompanied by the activation (induction of formation) of specific pro- or anti-apoptotic proteins, as well as specialized proteolytic enzymes—caspases. Among the factors triggering apoptosis, the formation of ROS during the “distorted” pathway of oxidative metabolism in the cell should be noted. Convincing evidence exists that mitochondria play a central role in ROS production and the subsequent development of apoptosis and necrosis. This involves changes in the permeability of mitochondrial membranes due to the formation of specific mitochondrial pore complexes and the initiation of mitoptosis (Figure 3) [175,176].

Under the influence of hydroxyl radicals, mitochondrial pores open, leading to the expression and release of pro-apoptotic proteins into the cytosol. The opening of these pores transforms mitochondria from “power plants” into “furnaces” for oxidation substrates without ATP production. Precise biochemical studies have established that disruptions in tissue oxygenation, hyperproduction of excitotoxic amino acids, a decrease in “normal” Ca^2+^ accumulation by mitochondria, and oxidative damage to mitochondrial membranes by ROS exacerbate pore opening and the release of apoptogenic proteins from damaged mitochondria.

In this context, the role of one neurotrophic factor—tumor necrosis factor-α (TNF-α)—is significant. TNF-α is associated with the opening of mitochondrial pores, subsequent membrane damage, and the progression of mitoptosis. The mitochondrial pore is a channel traversing both mitochondrial membranes and comprises three proteins: the adenine nucleotide translocator, the voltage-dependent anion channel (porin), and the benzodiazepine receptor. When this complex binds to Ca^2+^, substances with small molecular masses can pass through the membrane pore. This process leads to a decrease in membrane potential and matrix swelling, which inevitably compromises the integrity of the outer membrane. Consequently, apoptosis-related proteins are released from the intermembrane space into the cytoplasm [89,177].

There are several of them: apoptosis-inducing factor, secondary mitochondria-derived activator of caspases (Smac), and some procaspases. The inducing factor goes directly to the nucleus, where it causes DNA degradation. Along with specific apoptosis proteins, cytochrome C, which normally serves as the final link in the electrotransport chain, leaves the mitochondrion through the open pore. In the cytoplasm, this protein binds to the protein Apaf-1 (apoptotic protease activating factor-1) and forms an apoptosome complex. It, with the help of Smac and another factor (Omi/HtrA2), activates procaspase-9, which, becoming caspase-9, transforms two other proenzymes into caspases-3 and 7; and they already cleave structural proteins, leading to the appearance of biochemical and morphological signs of apoptosis [178,179,180]. Among the earliest events, in particular, are the translocation of phosphatidylserine to the outer membrane layer and DNA fragmentation under the influence of ROS and NO [181].

Among the secondary signs, the most characteristic are the “shedding” of the cell from the matrix, membrane wrinkling, nuclear condensation, and the formation of vesicles containing cellular contents—apoptotic bodies [182]. The release of cytochrome c into the cytoplasm is facilitated by a decrease in pH during the development of lactate acidosis and an increase in oxidative modification of mitochondrial proteins and lipids. This latter reaction is directly triggered by ROS, which are inevitably formed as a result of “parasitic” energy reactions [183]. Cytochrome C can be released in response to an increase in Ca^2+^ ion concentration, which triggers pore opening, and its release is also regulated by proteins of the Bcl-2 family [184,185].

They are the ones that regulate apoptosis at the mitochondrial level. Some of the proteins of this large family (Bcl-2, as well as Bcl-xL, Bcl-w, Mcl-1, Al, and Boo) prevent apoptosis; others (Bax, Bad, Bok, Bcl-xS, Bak, Bid, Bik, Bim, Krk, and Mtd) promote its initiation [186,187].

The entire superfamily of Bcl-2-related proteins is considered to be one of the most important classes of apoptosis-regulating gene products. Their ever-expanding list includes both cell death antagonists and apoptosis-inducing proteins. Due to the large number of representatives of this family described to date, they are usually divided into three groups [188,189].

Anti-apoptotic proteins Bcl-2, Bcl-xL, Bcl-w, Mcl-1, Bfl-1 and Boo with homology in the BH1, 2, 3 and 4 regions;Pro-apoptotic proteins Bax, Bak, Bad, Bok and Diva with homology in regions BH1, 2 and 3 but not BH4;“BH3-proteins” are pro-apoptotic proteins such as Bik, Bid, Bim, Hrk, and Blk, which share homology exclusively in the BH3 domain.

The combined action of related cell death agonists and antagonists from the Bcl-2 family represents a regulatory switch whose function is determined, at least in part, by selective protein-protein interactions. These proteins are characterized by the ability to form heterodimers in which partners repress each other [190].

Proapoptotic proteins are mostly localized in the cytosol and translocate to the mitochondrial membrane in response to certain stimuli. Some studies claim that Bach proteins translocate from the cytosol to the mitochondria, while others suggest that they undergo conformational changes enhanced by interaction with Bid proteins [191].

Bid proteins are known to be hydrolyzed by caspase-8 and their C-terminal part interacts with mitochondria. Several models have been proposed for the participation of Bcl-2 family proteins in the regulation of protein transfer from the mitochondrion to the cytosol [192,193,194].

Since Bax (like other Bcl-2 family proteins) can form pores in the outer mitochondrial membrane, it has been hypothesized that these pores might be large enough to allow cytochrome c to exit. However, this has not yet been confirmed;There is a hypothesis suggesting that Bax interacts with VDAC, resulting in the formation of an even larger channel capable of accommodating cytochrome C. Notably, the conductivity of this channel is blocked by Bcl-xL [195];Bcl-2 can also form channels in the outer mitochondrial membrane that allow adenine nucleotides to pass through. It is hypothesized that bach closes VDAC, ATP/ADP exchange between mitochondrion and cytoplasm is disrupted, resulting in the opening of RTR. Moreover, all these phenomena are prevented by Bcl-2 [196,197,198];There is also a hypothesis that suggests that the Bcl-2 family protein complex interacts with the giant pore complex in the inner mitochondrial membrane and leads to membrane depolarization, mitochondrial swelling, and cytochrome C release [199,200].

The Bcl-2 protein has a direct inhibitory effect on pore opening, but it does not protect against all permeability change inducers. Because the mitochondrial pore is regulated by a complex that includes Bcl-2 and Bax antagonists, changes in its stoichiometry (e.g., increased Bax synthesis or Bcl-2 modification) may contribute to permeability changes. It is hypothesized that the ratio between Bcl-2 and Bax proteins, and their phosphorylation, promotes either cell survival (excess Bcl-2 or Bcl-xL) or cell death (excess Bax, phosphorylation of Bcl-2). At the level of phosphorylation, there is a link between changes in mitochondrial permeability and receptor signaling pathways, as phosphorylation can be produced by protein kinases such as JNK (activated by various stress stimuli and through TNFR and Fas receptors, phosphorylates Bcl-2) and PKB/Akt, which transmits signals of growth factors NGF, IGF-1, phosphorylates Bad, preventing apoptosis [188,201,202,203,204,205,206].

Bcl-2 acts as a neuroantioxidant—it blocks the output of cytochrome C and prevents the development of apoptosis. Protein 53 (p53) takes part in triggering apoptosis caused by DNA damage, activation of oncogenes and hypoxia by interacting with Vach, stimulating “death receptors” and apoptosis genes. p53 activates the suicide modulator PUMA (p53 upregulated modulator of apoptosis), which then binds Bcl-2 and disables this protein that prevents apoptosis. Thus, the release of cytochrome C from mitochondria is no longer restrained by anything [112,207].

Some calcium ion-binding proteins, such as ALG-2, encoded by the gene of the same name (apoptosis-linked gene-2), also participate in the development of neuroapoptosis. Thus, the interaction between ALG-2 and the Alix protein (ALG-interacting protein X, also known as AIP1) regulates neuroapoptosis [208,209,210].

It is assumed that in addition to the participation of dysfunctional mitochondria in apoptosis processes, they play a key role in the cell’s choice of a pathway to realize the type of morphological death. Data from various researchers indicate that activation of the mechanisms of one or another form of cell death pathway may be determined by the number of open pores in dysfunctional mitochondria. If pores are formed in several mitochondria, the autophagy process is activated in the cell [24,211,212,213]. When pores open in more dysfunctional mitochondria, apoptosis is initiated in the cell, which is probably a consequence of an increase in cytochrome C and apoptosis-induced factor (AIF) in the cytoplasm. Eventually, when PT pores open in the cell in almost all dysfunctional mitochondria, dissociation of oxidation and phosphorylation and intensive ATP hydrolysis by mitochondrial ATPase occur, the mechanisms of necrosis-like cell death are activated [214,215,216,217]. In dysfunctional mitochondria the minimum number of open pores does not fundamentally affect the process of cell death, and with a larger number of open pores the initiation of apoptosis is possible, with generalized opening of mitochondrial pores the process of necrosis is realized [218,219,220]. The level of ATP production in dysfunctional mitochondria is of great importance in the “choice” between the realization of apoptosis and necrosis-like programmed cell death. At low ATP level in dysfunctional mitochondria the process of programmed cell death by necrosis mechanism proceeds, sufficient energy supply promotes the process of programmed cell death by apoptosis mechanism [183,221,222,223,224].

## 5. Possible Strategies for Pharmacocorrection of Mitochondrial Dysfunction

It is shown that the development of mitochondrial dysfunction is naturally accompanied by damage to brain matter [24,225,226]. The following ways of possible pharmacologic correction of mitochondrial dysfunction are currently postulated [6,227,228,229,230,231,232].

(1)increase in the efficiency of mitochondrial use of deficient oxygen due to prevention of dissociation of oxidation and phosphorylation, stabilization of mitochondrial membranes;(2)weakening of inhibition of Krebs cycle reactions, especially by maintaining the activity of the succinatoxidase link;(3)compensation of the lost components of the respiratory chain;(4)formation of artificial redox systems that shunt the electron-overloaded respiratory chain;(5)economization of oxygen utilization and reduction of oxygen demand of tissues or weakening of respiratory control in mitochondria, or inhibition of pathways of its consumption that are not necessary for emergency maintenance of vital activity in critical states (non-phosphorylating enzymatic oxidation—thermoregulatory, microsomal, etc., non-enzymatic oxidation of lipids);(6)increase in ATP formation during glycolysis without increasing lactate production;(7)reduction of ATP consumption by the cell for processes that do not determine emergency maintenance of vital activity in critical situations (various synthetic reductive reactions, functioning of energy-dependent transport systems, etc.);(8)introduction of high-energy compounds from outside.

Today, there is no drug that would influence all the above-mentioned ways of correcting energy metabolism. At present, to correct mitochondrial dysfunction and energy deficiency, mainly antihypoxants are used—agents that improve the body’s absorption of oxygen and reduce the brain’s need for it, thereby contributing to the body’s resistance to oxygen deficiency. From the biochemical point of view, hypoxia is a violation of substrate oxidation in the tissues of the body due to the impediment or block of electron transport in the respiratory chain, so the action of antihypoxants should be realized at the cellular level and be directed at the respiratory chain. To date, there is no single established classification of antihypoxants. This is due to the fact that the drugs are represented by compounds from different chemical classes and the mechanism of their action is not always studied. They can improve the oxygen transport function of blood or preserve the energy status of the cell under hypoxia. Direct energizing action of antihypoxants is aimed at correction of respiratory chain function under hypoxia conditions. In addition, there are antihypoxants of nonspecific action, the effects of which are aimed at the correction of functional-metabolic systems [233,234,235,236].

Currently, the mechanism of action of known antihypoxants is aimed at the following links of cell energy metabolism and includes drugs of different pharmacological groups:Electron carriers in the respiratory chain (coenzyme Q10 and its analog idebenone, succinic acid, vitamins K1, K3);cofactors of enzyme reactions of energy metabolism (nicotinamide, riboflavin, L-carnitine and others);correctors of lactate acidosis (dimefosfon).

Clinical and experimental studies of the last decade have established that in the acute period of ischemic stroke, pharmacological correction of energy supply should include restoration of electron-transport and conjugating function of the NAD-dependent part of the respiratory chain, as well as activation of compensatory metabolic mechanisms providing electron delivery to the cytochrome section of the chain [6,237,238,239].

### 5.1. Idebenone

Idebenone is an original drug of quinone structure, bearing a marked resemblance to natural CoQ _10_. Idebenone has reduced hydrophobicity, ability to penetrate the blood–brain barrier, activates the respiratory function of mitochondria and has a positive effect on the processes of free-radical oxidation in brain tissue. Under the conditions of in vitro experiment in nervous tissue culture, it was shown that idebenone prevents the formation of free radicals in cytosol and mitochondria, while in parallel reducing the concentration of marker products of oxidative modification of protein molecules. Idebenone is able to act as an electron carrier in the respiratory chain of mitochondria, increases the formation of ATP, and also increases glucose utilization in nervous tissue, in parallel, reducing the likelihood of lactate acidosis development [240,241,242]. In clinical trials, idebenone has demonstrated the ability to improve memory and learning ability in ischemic brain damage. The effectiveness of idebenone in multi-infarct dementia was confirmed by the results of multicenter studies. In cerebrovascular insufficiency of various degrees of severity, a course of therapy with idebenone at a dose of 90 mg/kg resulted in some improvement of cognitive functions after 1.5–2 months of treatment. The efficacy of idebenone in complex therapy of such primary mitochondrial dysfunctions as MELAS syndrome, Leber’s optic atrophy, and Ley’s disease has also been shown [243,244,245].

### 5.2. Menadione

Menadione or 2-methyl-1,4-naphthoquinone can be incorporated into the respiratory chain and by shunting the electron flow at the site from NADH KoQ (Coenzyme Q (CoQ)), restores the electron flow to cytochrome oxidase under hypoxia. Menadione administration to nerve cell cultures under conditions of moderate hypoxia has been shown to normalize NADH/NAD, increase mitochondrial respiration rate, and increase ATP concentration. Menadione regulates the Nrf2-dependent activation of gene expression followed by enhanced formation of protective proteins such as enzymes involved in glutathione biosynthesis [246,247,248]. Menadione was considered as a promising mitochondrial antioxidant and mitoprotector [6]. However, higher concentrations of menadione cause toxic oxidative stress associated with tissue damage, mitochondrial DNA damage, and cell death [249]. Menadione can undergo one-electron reduction, resulting in the formation of unstable free radicals that produce reactive oxygen species by rapid reaction with oxygen, thereby causing oxidative stress. However, other studies have shown that menadione is an effective inhibitor of lipid peroxidation in microsomes by suppressing lipid peroxide formation through various mechanisms, including binding to one-electron molecule transfer enzymes. According to other studies, reduced forms of menadione may exhibit antioxidant activity [250]. All of the above requires additional studies.

Synthetic ubiquinone-containing preparations from the class of redox polymers, among which Sodium polydihydroxyphenylene thiosulfonate-olifen (hypoxen) has gained popularity, contribute to the restoration of respiratory chain functioning.

### 5.3. Olifen

Olifen has a polyphenolic ubiquinone component in its structure, contributes to the reduction of electron leakage from the respiratory chain and exhibits antihypoxic and antioscidatic properties. Under conditions of oxygen deficiency, the use of oliphene is accompanied by oxidation of reduced nicotinic and flavinic nucleotides. It prevents the development of lipoperoxidation reactions of mitochondrial suspension membranes, preserves the charge of the outer mitochondrial membrane, and stimulates the destruction of peroxidation products [6]. Olifen is used in the manifestation of secondary mitochondrial dysfunction due to chronic myocardial ischemia and working hypoxia, due to physical overload. It shows actoprotective effect [251,252].

### 5.4. Succinic Acid

To enhance the alternative NADH-oxidase pathway of ATP formation, agents involved in succinatoxidase oxidation are used, the stimulation of which is achieved by activation of succinate dehydrogenase reaction through exposure to succinate-containing compounds that facilitate the transport of succinate into the cell. Succinate exerts its antihypoxic effect in two ways. First, it acts as a substrate of the tricarboxylic acid cycle and the enzyme succinate dehydrogenase. Second, it plays a role as a signaling molecule, activating HIF-1α and the orthologous receptors SUCNR1 and GPR91. Interaction with the latter promotes an increase in the level of reabsorbed glucose and stimulation of gluconeogenesis [253,254,255]. The combination of sodium succinate and cytochrome C is promising from the point of view of energetotropic and anti-ischemic action. Sodium salts of succinate are effective in reducing metabolic intracellular acidosis due to intracellular oxidation with the replacement of one hydrogen molecule by sodium to form bicarbonate. Antioxidant action of succinates is realized due to inhibition of production of reactive oxygen species by bioenergetic reactions of mitochondria. The antioxidant effect of succinate is demonstrated by a reduction in oxidative stress products, such as carbonylated proteins in the mitochondrial suspensions of the myocardium and brain in experimental animals. Succinate promotes the activation of endogenous antioxidant synthesis, specifically glutathione. It also stimulates erythropoiesis and increases the levels of adrenaline, noradrenaline, and dopamine, contributing to its psychostimulant, normothymic, and antidepressant effects. The efficacy of succinate-containing preparations has been demonstrated in cases of secondary mitochondrial dysfunction caused by myocardial ischemia, chronic cerebral ischemia, and work-induced hypoxia [256]. The possibility of using succinic acid preparations in primary mitochondrial dysfunction is actively discussed, particularly in a patient with MELAS syndrome [257,258]. Combination succinate-containing drugs such as reamberine, cytoflavin, and remaxol have found use in clinical practice [256,259,260,261]. The bioavailability of succinate is enhanced by combining it with various metabolites (such as citric and malic acids). Salts of succinic acid and mixtures, such as Limontar (a combination of sodium succinate and citric acid), are more readily accessible to mitochondria and are oxidized within them. A more promising approach today involves increasing the activity of succinate dehydrogenase and stimulating endogenous succinate production. This is achieved through pharmacological agents that act as succinate precursors and improve its penetration through the blood–brain barrier. However, such drugs have not yet found application in clinical medicine, and research in this area is currently limited to experimental studies [6,262].

Cytochrome C and KoQ preparations may be recommended in later stages of hypoxia as a redox mediator of the respiratory chain at the site between flavoprotein dehydrogenase and cytochromes stabilizes mitochondrial inner membranes, relieves succinate oxidase and NADH-oxidase inhibition [263,264,265].

In the 1960s–1970s, under the leadership of Academician V. Skulachev, a class of mitochondrial antioxidants was developed under the general name SkQ. To deliver antioxidants to mitochondria, a low-molecular-weight compound was proposed, consisting of a positively charged phosphorus atom surrounded by three hydrophobic phenyl groups (triphenylphosphonium, TPP, a charged triphenylphosphine). The mitochondrial protective properties of compounds combining TPP with ubiquinone and TPP with plastoquinone (a quinone involved in the electron transport chain during photosynthesis) were studied.

Currently, 11 compounds of this type are known: SkQ1, SkQR1, SkQ2, SkQ2M, SkQ3, SkQ4, SkQ5, SkQBerb, SkQPalm, C12TPP, and MitoQ. Plastiquinone derivatives with one, two, or three methyl groups are used as the antioxidant component. SkQR1 is considered the most active.

The mechanism of the mitochondrial protective action of such compounds involves their penetration into mitochondria and the reduction of ROS. This occurs both by decreasing ROS production during mitochondrial bioenergetic reactions and through direct interaction between SkQ compounds and ROS [27,266,267].

No less important drugs in conditions of cerebral hypoxia will be the so-called “correctors of lactate acidosis”. These medications include dimefosfon.

### 5.5. Dimefosfon

Dimefosfon is a dimethyl ester of 1,1-dimethyl-3-oxobutyl phosphonic acid. Dimefosfon had pronounced antihypoxic and anti-ischemic properties. The antihypoxic effect of dimefosfon was associated with a decrease in lactate production in brain structures simultaneously with an increase in the activity of key cellular energy supply enzymes: NADPH (diaphorase, succinate dehydrogenase and glycero-6-phosphate dehydrogenase), accompanied by an increase in ATP and creatine phosphate. Dimefosfon increased brain energy charge by decreasing oxygen consumption by brain tissue. Dimefosfon inhibits lipoperoxidation processes of mitochondrial meebras isolated from rat hippocampal neurons. It is active in relation to secondary mitochondrial dysfunction caused by cerebral ischemia. It is known to be a positive modulator of neuronal acetylcholoesterase [6].

However, despite the accumulated large clinical material on the use of the above-mentioned drugs, at present, they do not fully meet all the requirements for drugs correcting mitochondrial dysfunction. Thus, clear criteria and principles of their combined use, as well as the possibility of their use at different stages of bioenergetic hypoxia have not been developed. In addition, drugs are used chaotically, without sufficient knowledge about their capabilities, as well as without planning the treatment strategy from the position of expediency. In addition, a significant disadvantage of modern antihypoxants and neuroprotectors is their inability to affect the delicate molecular and biochemical chains of energy metabolism at already formed mitochondrial dysfunction, as well as their ineffectiveness with respect to the processes of cell death [6,217,268].

In this regard, another approach to correcting mitochondrial dysfunction is currently being considered—the use of antioxidant drugs that, by acting on the cell’s antioxidant system, reduce the concentration of cytotoxic marker products of oxidative destruction of proteins and nucleic acids. Through this effect, antioxidants can restore energy production in the mitochondrial respiratory chain and mitigate mitochondrial dysfunction, given the well-known role of ROS in its development.

However, it is important to emphasize that the use of antioxidant drugs is often inconsistent, and dosing regimens for these drugs are not clearly established. This can lead to a situation where antioxidants begin to act as pro-oxidants within cells.

In this context, so-called “thiol antioxidants,” which contain free SH-groups in their molecular structure, are of particular interest. According to several researchers, the presence of SH-groups allows thiol antioxidants to compete with ROS, forming “stable complexes” with them, thereby protecting the SH-groups in the cysteine-dependent region of mitochondrial inner membrane proteins. This action of antioxidants helps prevent the opening of mitochondrial pores under conditions of oxidative and nitrosative stress [6,269,270].

Recently, the thiol antioxidant N-acetylcysteine (NAC) has attracted the attention of pharmacologists and clinicians.

### 5.6. NAC (N-Acetylcysteine)

NAC acts as a “trap” for peroxynitrite and nitric oxide (NO), suppresses the production of IL-1β, and inhibits the activity of H_2_O_2_-dependent p38 stress kinases in astrocytes. It has been established that NAC, indirectly through the reduction of ROS levels, inhibits the functioning of the MAP kinase cascade, thereby decreasing the production of transcription factors. This, in turn, reduces the expression of genes responsible for the synthesis of NO synthase and COX-1 in astrocyte cultures [271,272,273].

It should be noted, however, that studies on the activity of NAC have been conducted predominantly in vitro and on model pathologies associated with brain ischemia, where it demonstrated relatively low therapeutic efficacy. Among other antioxidants, derivatives of hydroxypyridine—emoxypine and its succinate salt, mexidol—have found practical applications.

### 5.7. Emoxypine and Mexidol

Emoxypine and mexidol are highly effective inhibitors of free radical oxidation and suppress oxidative protein modification in ischemic brain tissue. Both drugs exhibit pronounced membrane-stabilizing effects (increasing the content of phosphatidylserine and phosphatidylinositol) and normalize the activity of various membrane-dependent enzymes, such as adenylate cyclase and creatine phosphokinase [6,274,275].

### 5.8. Ethylmethylhydroxypyridine Succinate

Ethylmethylhydroxypyridine succinate (Mexidol, Mexicor) is widely used today as an antioxidant and antihypoxant. It activates compensatory metabolic processes by facilitating succinate entry into the mitochondrial respiratory chain, thereby enhancing the energy-producing function of mitochondria, improving energy metabolism, and maintaining the levels of macroergic compounds under conditions of mitochondrial insufficiency.

Mexidol inhibits lipid peroxidation and increases the activity of antioxidant enzymes, such as superoxide dismutase and glutathione peroxidase, due to its 3-hydroxypyridine component. Additionally, Mexidol modulates the activity of membrane-associated enzymes (adenylate cyclase, acetylcholinesterase, etc.), neurotransmitter transport systems, ion channels, receptors, and receptor complexes (such as acetylcholine and GABA).

It also has the ability to reduce glutamate excitotoxicity and nitric oxide (NO) levels, while regulating the expression of early response genes such as c-fos and HIF-1mRNA [276,277].

Mexidol exerts a protective effect on the protein components of neuronal membranes, including receptors and ion channels, enhancing nerve conduction and synaptic transmission. It stimulates the energy-producing functions of mitochondria and ATP synthesis. Mexidol has been shown to improve the ultrastructure of myocardial mitochondria following ischemia and reperfusion.

In experimental studies, Mexidol alleviated cognitive deficits in animals during the recovery period after total brain ischemia, under conditions of emotional-pain stress, and following the administration of cholinomimetics. Mexidol demonstrated good therapeutic efficacy in the treatment of carotid stroke, promoting regression of neurological and cognitive deficits, normalizing EEG patterns, and showing no significant side effects [6,278].

Mexidol is used mainly in secondary mitochondrial dysfunctions accompanying chronic cerebral ischemia, CNS damage due to prenatal hypoxia, multiple sclerosis, and working hypoxia [259,279]. In primary mitochondrial dysfunctions, mexidol was used as part of complex therapy of MELAS syndrome and in mitochondrial myopathy [6,280].

### 5.9. Meldonium

Meldonium (mildronate) is a metabolitotropic cardioprotector. The main mechanism of its action is reversible inhibition of the rate of carnitine synthesis from its precursor—γ-butyrobetaine, which leads to a decrease in carnitine-mediated transport of long-chain fatty acids without changing the metabolic processes of short-chain fatty acids through mitochondrial membranes, i.e., there is no complete blockage of oxidation of all fatty acids [6,281]. Meldonium is used in secondary mitochondrial dysfunction due to myocardial ischemia, working hypoxia, cardiomyopathy after prenatal hypoxia, and cerebral ischemia [282,283,284]. There have been attempts to use meldonium for the treatment of primary mitochondrial dysfunction, particularly MELAS syndrome [6].

The thiol antioxidant and ROS/NO scavenger Thiotriazoline has gained wide international recognition. It was first synthesized in the USSR, specifically at the Zaporizhzhia Medical University in 1982, and became an original pharmaceutical drug in Ukraine in 1992.

Thiotriazoline enhances the energetic potential of the myocardium and mitigates oxidative stress in conditions such as ischemic heart disease, physical overexertion, and work-related hypoxia [6].

### 5.10. Thiotriazoline

Thiotriazoline (tiazotic acid (thiotriazoline)) The results of numerous studies have shown the ability of thiotriazoline to influence oxidative processes in mitochondria in ischemic lesions of the brain and heart (Figure 4). Thus, it was found that the anti-ischemic efficacy of the drug is based on its ability to reduce the degree of inhibition of oxidative processes in the Krebs cycle, activate the compensatory malate-aspartate shuttle mechanism and increase the production of ATP and ADP against the background of a decrease in AMP [285]. Under conditions of ischemic brain injury, Thiotriazoline normalizes glucose utilization within cells, increases the activity of glucose-6-phosphate dehydrogenase, restores the NAD/NADH ratio and cytochrome C oxidase activity, and elevates the levels of pyruvate, malate, isocitrate, and succinate. Simultaneously, it reduces lactate overproduction, mitigates uncompensated acidosis, and counters its pro-oxidant effects. Thiotriazoline is the only drug known to activate the conversion of lactate to pyruvate

In both in vitro and in vivo studies, Thiotriazoline inhibited mitochondrial ROS production. As a nitric oxide (NO) scavenger, it enhances NO bioavailability. Moreover, it prevents the irreversible inactivation of the NF-kappa B transcription factor by protecting cysteine residues—Cys 252, Cys 154, and Cys 61—in its DNA-binding domains from excessive ROS. Thiotriazoline also appears to facilitate the restoration of these residues during reversible inactivation, acting in the role of Redox Factor-1.

Preincubation of mitochondrial suspensions with Thiotriazoline (10^−5^ M) and subsequent addition of MPTP (1-methyl-4-phenyl-1,2,3,6-tetrahydropyridine) (60 μM), sodium nitroprusside (100 μM), or H_2_O_2_ (50 μM) significantly inhibited the rate of mitochondrial pore opening (*p* ≤ 0.05), increased mitochondrial membrane potential (*p* ≤ 0.05), and raised intramitochondrial GSH concentration (*p* ≤ 0.05).

Thiotriazoline preserves the threshold sensitivity of membrane receptors, maintains membrane fluidity, protects phospholipids from oxidation, prevents ion channel polarization, and normalizes ion transport. Additionally, Thiotriazoline inhibits NO-dependent apoptotic mechanisms and increases levels of the anti-apoptotic [6].

Our studies have established that tiotriazolin is effective in secondary mitochondrial dysfunction due to myocardial ischemia, cerebral ischemia, working hypoxia after exercise, and intrauterine hypoxia. It is also effective in congenital mitochondrial dysfunction leading to myocardial hypertrophy in children [149,286,287,288,289,290].

### 5.11. Angiolin

It has been established that in cases of already developed mitochondrial dysfunction and the initiation of apoptotic processes, metabolic therapy agents (such as coenzyme Q10, carnitine, B vitamins, and succinic acid derivatives) demonstrate limited efficacy and are unable to regulate the delicate aspects of energy metabolism for which they serve as intermediates (Figure 5).

Another approach to correcting mitochondrial dysfunction involves the use of thiol antioxidants. These agents compete with the SH-groups of cysteine-dependent sites on the inner mitochondrial membrane protein (ATP/ADP antiporter) for ROS and peroxynitrite, forming stable complexes with the latter. This mechanism prevents the opening of mitochondrial pores under conditions of oxidative and nitrosative stress.

This foundation led to the development of a fundamentally new metabolitotropic endothelial protector with an original structure—((S)-2,6-diaminohexanoic acid 3-methyl-1,2,4-triazolyl-5-thioacetate), named “Angiolin”. Angiolin exhibits anti-ischemic, cardioprotective, neuroprotective, and antioxidant properties [286,291,292,293].

The mitochondrial protective effect of Angiolin was established. Pre-incubation of the mitochondrial suspension with Angiolin (10^−5^ M) and the addition of MPTP (1-methyl-4-phenyl-1,2,3,6-tetrahydropyridine) (60 μM) significantly inhibited the rate of mitochondrial pore opening (*p* ≤ 0.05) and increased the mitochondrial membrane potential (*p* ≤ 0.05). Additionally, it elevated the intramitochondrial concentration of HSP70 (*p* ≤ 0.05) and reduced (*p* ≤ 0.05) the number of damaged mitochondria compared to control samples [6].

Angiolin (100 mg/kg) is effective in secondary mitochondrial dysfunction following chronic cerebral ischemia and intrauterine hypoxia. Angiolin significantly reduced the number of damaged mitochondria in CA1 hippocampal neurons and increased the intramitochondrial concentrations of HSP70 and GSH [291,294,295].

### 5.12. Benzodiazepines

Recent studies have shown that the manifestations of mitochondrial dysfunction in CNS ischemic lesions can be reduced by regulating the opening of mitochondrial pores. Given that the mitochondrial pore contains a regulatory benzodiazepine receptor, benzodiazepine receptor modulators deserve attention. Among them, the derivatives of 1,4-benzodiazepine and 1,3,4-benzotriazepine are of the greatest interest, as they have a wide range of neurotropic activities: antidepressant, antihypoxic, and nootropic effects. However, the effect of these drugs on mitochondrial functional activity during brain ischemia has not been studied, which defines the potential for further research in this area [296,297,298]. Pre-incubation of the mitochondrial suspension with a benzodiazepine receptor modulator (cinazepam, 10^−5^ M) and the addition of MPTP (1-methyl-4-phenyl-1,2,3,6-tetrahydropyridine) (60 μM) showed a weak inhibitory effect on the rate of mitochondrial pore opening, increased the mitochondrial membrane potential, and slightly increased the intramitochondrial concentration of HSP70. However, it did not significantly affect the number of damaged mitochondria compared to the control samples [6].

### 5.13. Estrogens and Selective Estrogen Receptor Modulators

It is known that, alongside their effects on the reproductive system, estrogens have a multifaceted impact on higher brain functions. It has been shown that estradiol plays a key role in the prenatal and early postnatal development of the brain, as well as throughout life in various disorders of the hypothalamic-pituitary-ovarian system. The main mechanism of estrogen action is genomic, which occurs at the level of the cell nucleus; the non-genomic mechanism exerts its effect through the cell membrane. To implement genomic effects in the brain, there are two types of receptors—estrogen receptor α (ERα) and estrogen receptor β (ERβ).

Tamoxifen and livial, which are anti-estrogens for breast tissue, exert their anti-estrogenic action through the activation of corepressors. Conversely, tamoxifen and livial, being agonists for endometrial cells, exhibit estrogenic effects through the stimulation of coactivators, while livial has no effect on endometrial cells [299,300,301,302].

Different concentrations of adapter proteins in estrogen-sensitive cells determine the selectivity and agonist/antagonist nature of tamoxifen and livial. It has been established that tamoxifen and its metabolite can inhibit the neurotoxic glutamate system and exert neuroprotective effects in β-amyloid-induced amnesia [303,304]. Experimental studies have established a rather high neuroprotective efficiency of estrogen receptor modulators in conditions of deprivation of the glutathione link of the thiol-disulfide system in the suspension of isolated neurons. Such an effect of estrogen receptor modulators, in our opinion, is explained, first, by their direct antioxidant effects. Secondly, it is known that estrogen receptor activation involves uncoupling of estrogen receptors from heat shock proteins (hsp), which ensures the penetration of HSP70 inside the cell.

Thirdly, it has been established that selective estrogen receptor modulators modulate the expression of global transcription factors, particularly AP-1, which is responsible for the synthesis of key enzymes in the antioxidant and thiol-disulfide systems. This explains the significant ability of tamoxifen and livial to restore the activity of superoxide dismutase, glutathione S-transferase, and glutathione reductase. Additionally, the anti-apoptotic effect of estrogens has been noted, which is due to the stimulation of the expression of anti-apoptotic proteins from the Bcl-xL family. Thus, our results reveal the importance of the neuronal glutathione system as a key target for neuroprotective therapy and provide experimental justification for the clinical use of estrogen receptor modulators—tamoxifen and livial—as neuroprotective agents. Tamoxifen is effective in secondary mitochondrial dysfunctions resulting from cerebral ischemia. Our experimental studies have established the neuroprotective activity of the selective estrogen receptor modulator tamoxifen citrate, which occurs, firstly, due to its ability to increase the levels of Hsp70 proteins in the brain tissue during the acute phase of ischemia. Secondly, tamoxifen citrate is capable of limiting the development of oxidative and nitrosative stresses, leading to a reduction in the concentrations of homocysteine and nitrotyrosine in the brain and an increase in glutathione, thus restoring thiol-disulfide balance in nerve cells. The synergistic enhancement of these effects of tamoxifen citrate under acute cerebral ischemia conditions led to a pronounced neuroprotective effect—reducing mortality and neurological deficit [71,305].

### 5.14. Neuropeptides

Currently, the search for neuroprotective and energy-tropic drugs of HSP-mediated action is conducted among neuropeptides capable of increasing HIF-protein concentration by modulating the expression of global transcription factors [149].

The discovery of neurotrophic peptide factors prompted the formation of a new strategy of pharmacotherapy-peptidergic, or neurotrophic therapy of neurodegenerative diseases [306,307]. A new class of neuroprotective agents has been created on the basis of peptide preparations. The ideology of creating these drugs is based on the recognition of the role of neuropeptides as universal “integrators” that unite and coordinate the activities of the three main regulatory systems of the organism—nervous, endocrine and immune systems [308].

A number of drugs with neurotrophic properties have been developed that are successfully used in the therapy of a wide range of neurological disorders. Cerebrolysin, Cerebrocurin, Cortexin, and Semax, which have been successfully used in the clinic of neurological disorders for ten years already, have been the most successful (Figure 6) [6,309]. Among the presented drugs, a special place is occupied by the domestic neuropeptide drug—cerebrocurin, which includes amino acids, neuropeptides, as well as low-molecular-weight products of controlled proteolysis of proteins and peptides from cattle embryos. The mechanism of action and targets of cerebrocurin are fundamentally different from other neuropeptide drugs, particularly from cerebrolisin. Cerebrocurin contains peptides that carry a program for analyzing the state and construction of the CNS. Thus, the final effect differs due to the qualitatively different mechanism of action. In addition, cerebrocurin increases the affinity of BDNF for its receptors. When cerebrocurin was administered to animals with cerebral ischemia, an increase in ATP production in oxidative reactions in the brain (cortex, CA1 hippocampus) was observed, as evidenced by an increase in malate content, enhanced activity of mitochondrial malate dehydrogenase, and cytochrome C oxidase. Cerebrocurin not only affected energy production but also its transport and utilization, as indicated by the increase in the activity of mitochondrial and cytoplasmic creatine phosphokinase. An important aspect of cerebrocurin’s effect on energy metabolism under brain ischemia conditions was the significant reduction in lactate production and lactate acidosis. Through the regulation of early response gene expression (c-fos) and the anti-apoptotic protein bcl-2, cerebrocurin is able to influence neuroapoptosis processes to some extent in cerebral ischemia conditions. The reduction of mitochondrial dysfunction in the context of ischemic brain pathology and the normalization of neuronal energy metabolism upon cerebrocurin administration contributed to the preservation of the main morphofunctional characteristics of neurons in the sensorimotor cortex during cerebral ischemia. Pre-incubation of the mitochondrial suspension with cerebrocurin and the addition of MPTP (1-methyl-4-phenyl-1,2,3,6-tetrahidropyridine) (60 μM) significantly inhibited the rate of mitochondrial pore opening, increased the mitochondrial membrane potential, and elevated the intramitochondrial concentration of HSP70. Moreover, it had a significant effect on the number of damaged mitochondria compared to control samples [6,310].

Experimental studies have revealed that cerebrocurin prevents microglia hyperactivation and reduces the production of IL-1α and other pro-inflammatory cytokines, reflecting the drug’s impact on the severity of local inflammatory reactions and processes. Cerebrocurin is effective for pharmacocorrection of both primary mitochondrial dysfunction (MELAS, MERRF) and secondary mitochondrial dysfunction, especially as a result of cerebral ischemia, intrauterine hypoxia, and ischemic opticopathy [311,312,313,314]. These neuropeptide drugs have also been successfully utilized in the management of various comorbid conditions, including hypertension, thyroid pathology, alterations in microbiota, chronic kidney disease, pregnancy, osteoarthritis, and type 2 diabetes mellitus [315,316,317,318,319,320]. Their broad therapeutic potential highlights their applicability across a diverse spectrum of systemic and metabolic disorders (Table 1) [321,322,323,324,325].

## 6. Conclusions

In summary and synthesizing the literature on the research topic, it can be concluded that antioxidants, selective modulators of estrogen and benzodiazepine receptors, and neuropeptide drugs are increasingly being applied both in model pathologies and in clinical neurology. At the same time, a number of issues related to the neuro- and energetropic effects, mechanisms of action of these drug groups, and especially Thiotriazoline, Cerebrocurin, and the new original drug “Angiolin,” require more detailed and in-depth study. All of this underscores the need to investigate the effects of the aforementioned drugs on mitochondrial dysfunction, energy, and energy-dependent processes occurring in brain cells during ischemic injury.

## Figures and Tables

**Figure 1 antioxidants-14-00108-f001:**
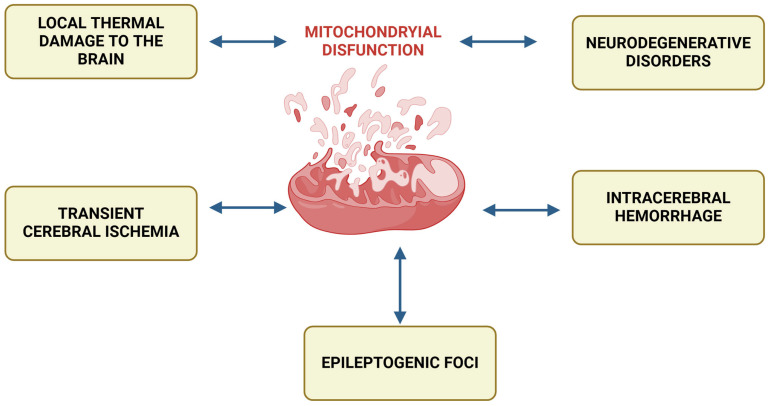
Diseases associated with mitochondrial dysfunction in cellular energy metabolism. The Figure was designed using BioRender.com. Mitochondrial dysfunction is a key factor in the pathogenesis of many central nervous system diseases and cerebrovascular pathologies, including intracerebral hemorrhage, epileptogenic seizures, localized thermal brain injury, neurodegenerative disorders, transient cerebral ischemia, chronic fatigue syndrome, migraines, alcoholic encephalopathy, senile dementia, and neuroinfections [6]. Currently, two types of mitochondrial dysfunction are recognized: primary, resulting from a congenital genetic defect, and secondary, arising from various pathological factors such as hypoxia, ischemia, oxidative and nitrosative stress, and the expression of pro-inflammatory cytokines. In modern medicine, increasing importance is being placed on the study of systemic disturbances in cellular energy metabolism, known as mitochondrial pathology or mitochondrial dysfunction.

**Figure 2 antioxidants-14-00108-f002:**
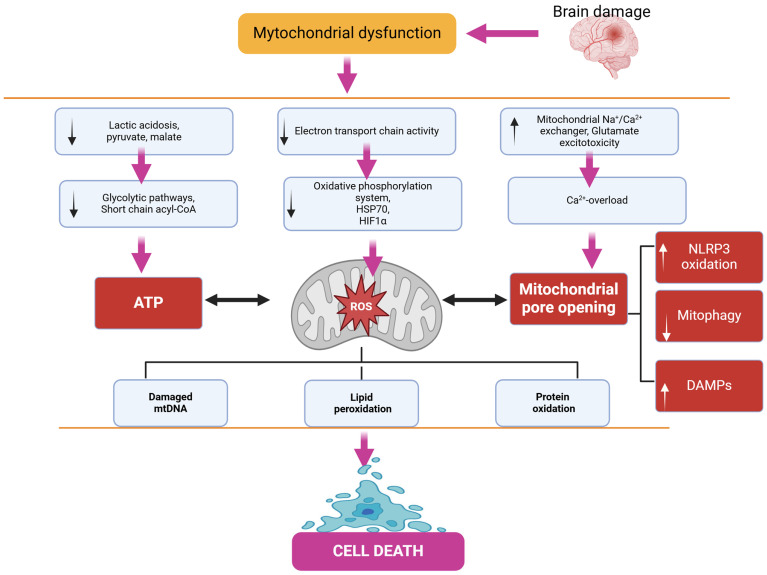
Diseases associated with primary mitochondrial dysfunction. The Figure was designed using BioRender.com. Cerebral ischemia, resulting from a sharp decrease in oxygen partial pressure (pO2), leads to discoordination in the Krebs cycle, inhibition of compensatory energy shunts (such as the Roberts shunt and the malate-aspartate shuttle), activation of glycolysis, lactic acidosis, phospholipase activation, and disruption of calcium (Ca^++^) transport. ATP production impairment and energy deficit contribute to glutamate excitotoxicity, Ca^++^ overload, activation of neuronal nitric oxide synthase (nNOS), and excessive nitric oxide (NO) production. Increased ROS production in mitochondria through NAD(P)H-dependent reactions, combined with excess NO, triggers bursts of free radical reactions. In the context of endogenous antioxidant deficiency, this leads to oxidative and nitrosative stress and reduced expression of HSP70 and HIF proteins, impairing mitochondrial functional activity. Energy production in mitochondria during ischemia relies on the functioning of the malate-aspartate shuttle, regulated by HSP and HIF proteins. Oxidative and nitrosative stress, driven by a significant shift in the thiol-disulfide balance and accumulation of cytotoxic nitric oxide derivatives (peroxynitrite, nitroxyl, nitrosonium), results in oxidative inhibition of mitochondrial respiratory chain enzymes and direct cytotoxic modifications of mitochondrial proteins and membrane lipids. The enhancement of free radical production under acidosis is linked to the increased release of iron, a trigger for oxidative mechanisms, from transferrin-like proteins in an acidic environment, intensifying Haber-Weiss reactions. Disruption of electron transport in mitochondria leads to secondary ROS generation, further amplifying oxidative and nitrosative stresses. ROS and free radicals cause oxidative modification of mitochondrial protein structures, particularly pore proteins, increase membrane permeability, and impair translation and transcription processes, as well as protein synthesis and import. Ultimately, damaged mitochondria initiate cell death programs via apoptosis or necrosis. Arrows indicate increase and decrease.

**Figure 3 antioxidants-14-00108-f003:**
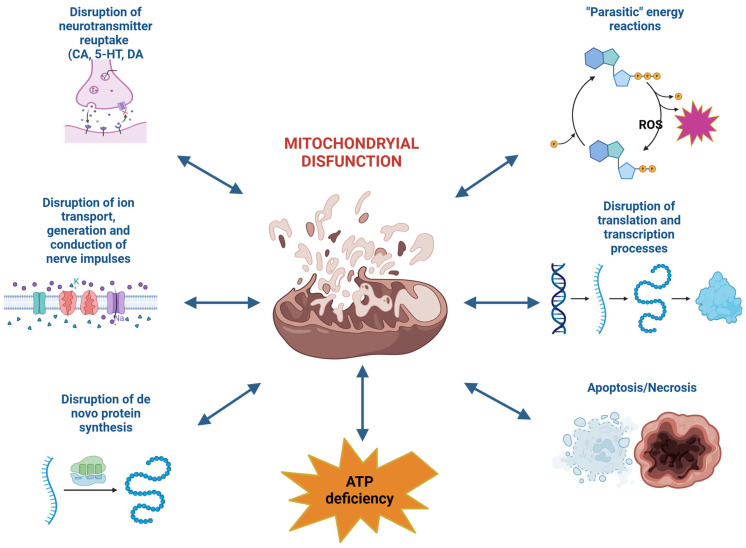
Disruptions in mitochondrial function. The Figure was designed using BioRender.com. Mitochondrial dysfunction is a typical pathological process lacking etiological and nosological specificity, ultimately leading to cell death. It results in energy deficit, impaired neurotransmitter reuptake, neurotransmitter autocoidosis, disrupted Ca^++^ transport, impaired nerve impulse conduction, and hyperproduction of ROS by bioenergetic systems. ROS oxidize the thiol groups of the Cys-dependent region of the inner mitochondrial membrane protein (ATP/ADP antiporter), causing the massive release of pro-inflammatory and pro-apoptotic factors. In the context of depleted mitochondrial antioxidant defenses (MnSOD, GSH), ROS induce oxidative modification of proteins, nucleic acids, and lipids.

**Figure 4 antioxidants-14-00108-f004:**
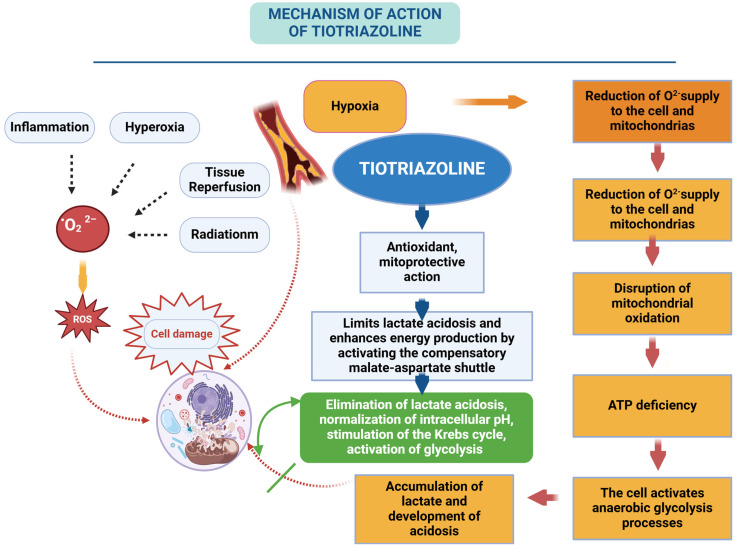
Action mechanism of Thiotriazoline. The Figure was designed using BioRender.com. The antioxidant properties of thiotriazoline, particularly its ability to act as a scavenger of ROS and NO, can be attributed to the reactivity of the sulfur atom in the morpholine thioazotate molecule. Thiotriazoline reduces ROS production by mitochondrial bioenergetic reactions and can influence the thiol-disulfide balance under ischemic conditions, protecting macromolecules (proteins, nucleic acids, and lipids) from oxidative modification. It inhibits lipid peroxidation of membrane phospholipids, normalizing the physicochemical parameters of membrane structure: structural integrity, free radical quenching rate, and microviscosity. Thiotriazoline increases ATP levels during ischemia and hypoxia by normalizing the Krebs cycle and activates the compensatory malate-aspartate shuttle under conditions of subtotal ischemia. It exhibits mitochondria-protective effects, enhances the utilization of glucose, free fatty acids, and glycogen, and reduces lactic acidosis. Thiotriazoline is the only drug that activates the conversion of lactate to pyruvate [6].

**Figure 5 antioxidants-14-00108-f005:**
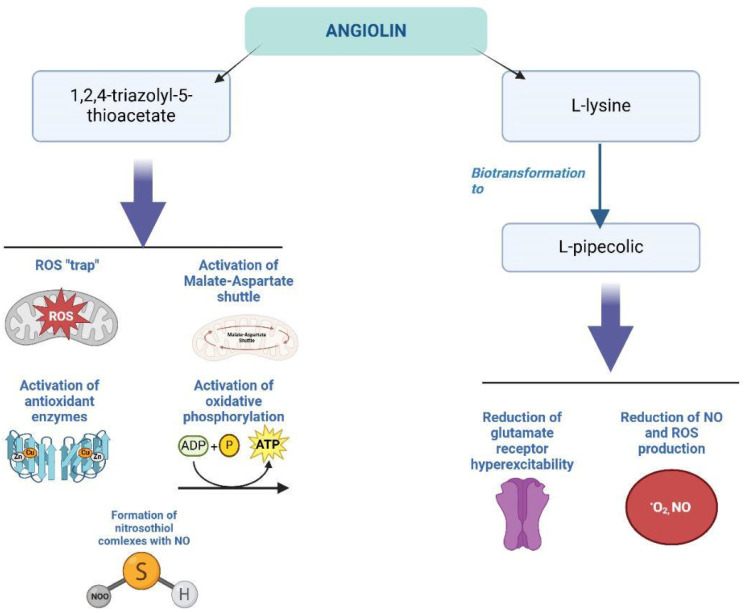
Action mechanism of Angiolin. The Figure was designed using BioRender.com. Angiolin exhibits anti-ischemic, antioxidant, and neuroprotective properties. The L-lysine component of the Angiolin molecule can be transformed in the body into pipecolic acid, which increases the affinity of the GABA-benzodiazepine-receptor complex and reduces excitotoxicity, thereby providing neuroprotective effects. Angiolin demonstrates anti-ischemic action by activating a compensatory ATP production mechanism (the malate-aspartate shuttle). Its antioxidant properties are due to its scavenging ability against cytotoxic forms of NO and ROS. The NO-scavenging properties are realized through the reactivity of both the cationic and anionic parts of the 3-methyl-1,2,4-triazolyl-5-thioacetate (S)-2,6-diaminohexanoic acid molecule. Specifically, L-lysine interacts with NO via its ε-amino group, resulting in the formation of the corresponding N-nitrosylated derivative. The ability of both the cationic and anionic parts of the Angiolin molecule to act as NO scavengers endows it with remarkable antioxidant properties. Angiolin inhibits NO-dependent neuroapoptosis mechanisms and regulates ROS-dependent mitochondrial pore mechanisms. Due to its antioxidant properties, Angiolin can influence ROS and SH-SS-dependent mechanisms of redox regulation and transcription, potentially leading to increased HSP70 expression.

**Figure 6 antioxidants-14-00108-f006:**
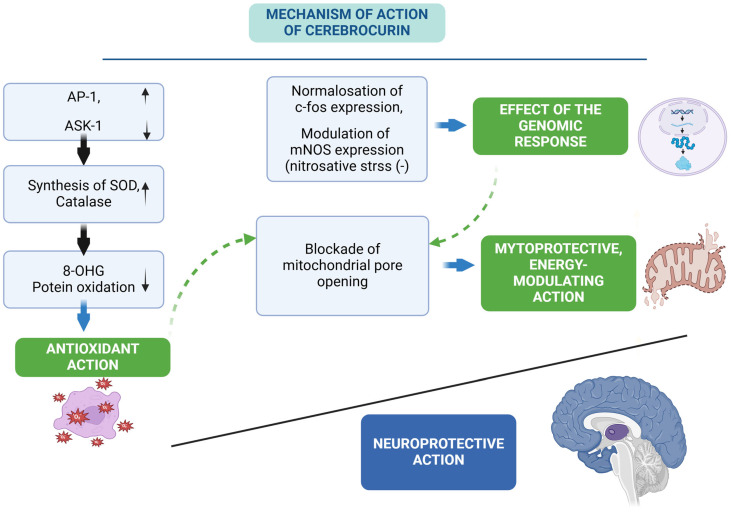
Action mechanism of Cerebrocurin. The Figure was designed using BioRender.com. The molecular basis of the multifunctionality of Cerebrocurin (a complex of free amino acids, peptides, and low-molecular-weight products of controlled proteolysis from the brain proteins of bovine embryos) is linked to the presence of multiple ligand-binding groups designed for different cellular receptors. Cerebrocurin contains peptides that carry information for the analysis of the state and construction of the central nervous system (CNS). It can perform a mediator function, modulate the reactivity of specific groups of neurons, stimulate or inhibit the release of hormones and pro- and anti-inflammatory cytokines, regulate tissue metabolism, or act as an effector of physiologically active agents. Cerebrocurin optimizes energy metabolism, intracellular protein synthesis, slows down the processes of glutamate-calcium cascade and oxidative stress, enhances the expression of Cu-Zn-SOD and catalase, and reduces the oxidative modification of nucleic acids and proteins during cerebral ischemia. The drug exhibits neurotrophic effects, protecting the neuronal cytoskeleton by inhibiting calcium-dependent proteases, including calpain, and increasing the expression of microtubule-associated protein 2 (MAP2) and the binding affinity of BDNF to its receptors. In acute cerebral ischemia, Cerebrocurin can modulate the expression of the early response gene c-fos, activate transcription factors, and initiate the synthesis program of adaptive proteins (HSP70 and HIF-1), anti-apoptotic proteins, and antioxidant proteins (SOD) in neurons. One of the key mechanisms of Cerebrocurin’s neuroprotective action is the inhibition of neuroapoptosis.

**Table 1 antioxidants-14-00108-t001:** Pharmacological characterization of agents with potential protective effects on mitochondria.

Name	Action Type	Dose/Concentration	Effects
Idebenone 2-(10-Hydroxydecyl)-5,6-dimethoxy-3-methylcyclohexa-2,5-diene-1,4-dione	Antioxidant	90 mg/kg	Improvement of neurological and cognitive functions in patients with cerebral underactivity. Effective in the therapy of primary mitochondrial dysfunctions (MELAS syndrome, Leber’s optic atrophy, Ley’s disease) [243,244,245].
		10^−5^ M	In neuronal culture prevents the formation of ROS in mitochondria. Inhibits neuroapoptosis. Electron carrier in the mitochondrial respiratory chain, increases ATP, reduces lactate acidosis [326,327].
Menadione (Vitamin K3) is a synthetic analogue of 1,4-naphthoquinone with a methyl group in the 2-position	Anti-/prooxidant	10^−7^–10^−5^ M	In vitro inhibits the generation of ROS, a membranostabilizer. Prevents electron leakage. In neuronal culture, it reduces the degree of functional disorders of mitochondria. However, there is also data on its pro-oxidative properties [246,247,248,328].
Olifen (Hypoxen), the sodium salt of poly-(2,5-dihydroxyphenylene)-4-thiosulfonic acid	Antioxidant, antihypoxant	50–100 mg/kg	Inhibits lipoperoxidation of mitochondrial membranes, preserves the charge of mitochondrial outer membrane in the brain of rats with cerebral ischemia [6].
		0.5–1.5 g	During physical exertion increases aerobic oxidation of energy substrates, increases work capacity [251,252].
		0.25–0.5 g	Improved neurological changes in alcoholism. Increased the body’s resistance to mixed hypercapnia and hypoxia. Increasing work capacity and resistance of the organism to psychophysiological loads [329].
Succinic acid	Antihypoxant	100–250 mg/kg	Acts as a substrate of the tricarboxylic acid cycle and the enzyme succinate dehydrogenase, and also plays the role of a signaling molecule, activates HIF-1α and the orphan receptors SUCNR1 and GPR91. Sodium succinate and preparations containing it (reamberin, cytoflavin, remaxol) increased ATP production by mitochondria in cerebral ischemia, working hypoxia [6,254,256].
		100–500 mg	Administration to patients with succinate for MELAS produced a persistent effect without ischemic stroke resulting in regression of local and general cerebral symptoms. Administration of recurrent stroke-like episodes. Cytoflavin in the first hours of stroke preserved brain matter, reduced neurologic deficit, and improved daily activity of patients [6,258].
SkQR1 10-(6′-Plastoquinonyl) decyltriphenylphosphonium	Mitochondrial antioxidant	100–250 nmol/kg	Mitoprotective action consists of proinduction into mitochondria and reduction of ROS both by inhibition of production by bioenergetic reactions of mitochondria and by direct interaction with ROS. Reduced neurological deficit, the number of damaged mitochondria, prevented accumulation of phosphorylated histone H2AX (γ-H2AX) in rats with cerebral ischemia [6,27,266,267,330,331].
Dimephosphon-dimethyl ester of 1,1-dimethyl-3-oxobutyl phosphonic acid	Antihypoxant	100–500 mg/kg	In cerebral ischemia in rats decreased lactate production, increased diaphorase, succinate dehydrogenase and glycero-6-phosphate dehydrogenase activity, increased ATP and creatine phosphate. Increased mitochondrial O_2_ consumption at mitochondrial respiration rates not due to impaired coupling of respiration and phosphorylation [6].
N-acetylcysteine	Antioxidant	10^−5^ M	Scavenger of NO, inhibits nitrosative stress, suppresses IL-1β production, p38-stresskinase activity in astrocytes and neurons in vitro. Reduces oxidative damage to mitochondrial membranes in vitro [6,271,272,273].
Emoxypine (methylethylpyridinol hydrochloride) and Mexidol (methylethylpyridinol succinate)	Antioxidants and antihypoxants	100–250 mg/kg	Reduce lethality and neurological deficit in animals with cerebral ischemia, increase ATP in mitochondria, activate aerobic oxidation processes, reduce oxidative damage to mitochondria. Reduce ultrastructural damage to brain mitochondria. Mexidol has a more pronounced mitoprotective effect. Mexidol reduces survival and neurological impairment after intrauterine hypoxia, improves the energy function of mitochondria [6,274,275,276,277].
		200–500 mg	Mexidol in the first hours after ischemic stroke improves quality of life. Mexidol has been used as part of complex therapy of MELAS syndrome and in mitochondrial myopathy improves motor and cognitive functions [6,280].
Meldonium (Mildronate), 2-(2-carboxyethyl)-1,1,1-trimethylhydrazinium	antihypoxant	100–250 mg/kg	Under conditions of ischemia or hypoxia, slows down the process of fatty acid transport across mitochondrial membranes, activates glycolysis. Reduced the formation of ROS by mitochondria [6,282,283,284].
Thiotriazoline, morpholinium thiazotate	Antioxidant, anti-ischemic agent	10^−7^–10^−5^ M 50–100 mg/kg	Reduced oxidative damage of mitochondria in vitro. In cerebral ischemia in rats it reduced lethality, normalized energy metabolism. Reduced the formation of ROS by mitochondria. Effective in secondary mitochondrial dysfunction, cerebral ischemia and after intrauterine hypoxia [149,286,287,288,289,290].
Angiolin, (S)-2,6-diaminohexanoic acid 3-methyl-1,2,4-triazolyl-5-thioacetate	Antioxidant, anti-ischemic agent	10^−7^–10^−5^ M 50–100 mg/kg	In vitro in mitochondria suspension reduced mitochondria damage and increased HSP70 concentration in them. Reduced animal death, neurological disorders, reduced mitochondria ultrastructural damage. Increased ATP in mitochondria, activated malate-aspartate shuttle mechanism, normalized eNOS/iNOS ratio and thiol-disulfide system [291,294,295].
Benzodiazepines	Mitochondrial pore regulation	10^−5^ M	In vitro, mitochondrial damage was reduced in mitochondrial suspension [6,296,297,298,332].
Selective estrogen receptor modulators (Tamoxifen, Livial)	Antioxidants	10^−5^ M 1 mg/kg	In vitro in mitochondrial suspension reduced mitochondrial damage, increased HSP70 in mitochondria. In conditions of cerebral ischemia, reduced oxidative stress, increased the activity of glutathione system [6,71,303,304,305].
Cerebrocurin	Neurotrophic cerebro-protector	150 µL/kg	In vitro in mitochondrial suspension reduced mitochondrial damage and increased HSP70 in mitochondria. Reduced animal death, neurological disorders, neuroinflammation and neuroapoptosis. Effective in primary mitochondrial dysfunction (MELAS, MERRF) and secondary dysfunction, especially due to cerebral ischemia, intrauterine hypoxia and ischemic opticopathy [6,311,312,313,314,321,322,323,324,325].

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
