# Peer review of "Targeting Mitochondrial Dysfunction in Cerebral Ischemia: Advances in Pharmacological Interventions"

_antioxidants, 2025, doi:10.3390/antiox14010108_

Round 1
Reviewer 1 Report
This review manuscript provides new information on mitochondrial dysfunction in pathological conditions, primarily pharmacological correction. The title is specific to cerebral ischemia, but the content does not focus on it. Thus, the authors should revise either the title or the content. The nerve is a bundle of axons that exits the brain and spinal cord. It is not the best word to use in the title.
Cerebral ischemia includes focal and global ischemia, which are different pathological conditions. The authors should cite the articles studying mitochondrial dysfunction in these patients or animal models of middle cerebral artery occlusion, forebrain ischemia, cardiac arrest, etc.
What kind of mitochondrial dysfunction biomarkers can be detected in stroke patients? When will they increase after stroke onset? Is mitochondrial dysfunction reversible? These questions have not been answered in this review manuscript.
The authors list 14 possible strategies for pharmacological correction of mitochondrial dysfunction. It would be ideal to provide a table showing which disease models these were tested in and whether clinical trials have been performed on stroke patients.
It would be ideal to provide a table showing which disease models 14 pharmacological corrections of mitochondrial dysfunction were tested in and whether clinical trials have been performed on stroke patients.
Author Response
We thank the reviewer for the evaluation of our manuscript and constructive feedback.
1. We now adhere to a clear narrative structure focused on cerebral ischemia in our manuscript. Any deviations are minimal and solely for clarifying the mechanisms involved in the progression of events.
2. We have revised the title and corrected stylistic errors in the narrative.
3. Cerebral ischemia can be local or partial, depending on the etiology or experimental model. However, they share common pathogenetic mechanisms, such as excitotoxicity, energy deficiency, oxidative stress, etc., leading to neuronal death. Each case is now appropriately referenced.
4. We discuss the most well-known markers of mitochondrial dysfunction and their informational value in this manuscript. Detailed information on molecular, genetic, biochemical, and histological markers and their dynamics is indeed very informative but will be covered in our subsequent works.
5. We have now taken reviewer's suggestions into account and compiled a summary table of pharmacological agents affecting mitochondrial dysfunction.
Reviewer 2 Report
Overall, I find this review confusing. The main reason is that this review does not work out a definition what mitochondrial dysfunction is in the context of this manuscript. Generally, any kind of pathological but also physiological stress can lead to dysfunctional mitochondria. Cells have mechanism in place to remove not functional mitochondria. There are most likely significant organ specific differences, as the brain consists of mostly non-dividing cells, but the liver has a high tissue turn-over. For example, inducing opening of the PTP is much harder in brain mitochondria than in liver mitochondria. Many references used in this manuscript are not specific for neurons or the brain. The title of this manuscript “Formation of mitochondrial dysfunction of nerve cells in cerebral ischemia and possibilities of pharmacological correction” does not fit the manuscript, because many references and contents are not specific to the brain and are even less specific to cerebral ischemia.
Another reason why this paper is difficult to read is, it contains a lot of detailed information, but I am missing a clear line. Some content is repeated multiple times, like mitochondria as the main source for reactive oxygen species. This review will benefit a lot if it focusses only on the role of mitochondrial dysfunction in the brain or during ischemic events to the brain. Considering the title, I would suggest for chapter 5 (Possible strategies for pharmacocorrection of mitochondrial dysfunction) to remove all drugs that have a detrimental effect on mitochondria (for example menadione). I would also suggest to change the word “formation” in the title with “onset”, because the onset/beginning of any mitochondrial dysfunction would have the biggest potential for treatment.
Line 126: mitochondrial pores: meant is the permeability transition pore. It is an accepted term and should be used. Same for line 449.
Line 129: Not only the ADP/ATP translocase can switch into the permeability transition pore, but the ATP synthase so too. Please see (Neginskaya et al 2022, Karch et al 2019)
Line 133: Replace “giant pores” with permeability pores.
Line 134: Molecules up to 1,500 Da are normally released.
Line 154: replace “is” with “are”
Line 171: As far as I am aware, there are no cure for the in Figure 1 depicted diseases, that would re-establish mitochondrial function. Please change "availability" to "unavailable".
Line 208-210: Please provide an explanation and reference why pyruvate dehydrogenase and malate dehydrogenase are markers of mitochondrial dysfunction. The abstract of reference 49 says that “AMA-M2 reacted with the pyruvate dehydrogenase complex-E2, branched-chain 2-oxo acid dehydrogenase complex and 2-oxoglutaric acid dehydrogenase complex in the assay utilized for this study” . I think these 2 papers are incorrectly referenced.
Line 340-342: research has shown that this model of the permeability transition pore is not any longer correct. Current models of the mPTP are the ADP/ATP translocase without porin/VDAC, the benzoediazepin receptor or any energy transmitting or consuming kinases like hexokinase or creatine kinase. Please see “Both ANT and ATPase are essential for mitochondrial permeability transition but not depolarization” by Neginskaya Morris and Pavlov in iScience 2022. Or Karch et al “Inhibition of mitochondrial permeability transition by deletion of the ANT family and CypD” in Sci. Adv., 5 (2019).
Line 679: Please add a reference for the introduction of the term “neuroapoptosis”
Line 816-838: In line 820 are several references listed, but it is not clear, whether they are in support of the bullet points 1-8. Please match references and bullet points.
Line 867: Typo “Idebeon” should be idebenone. It is not a derivative of ubiquinone, but a synthetic analog of ubichinone/coenzyme Q.
Line 884: Menadione; all I find about this drug is detrimental to the function of mitochondria. Please revise this paragraph.
Line 905: Olifen; did the authors mean oleficin? Oleficin is a non-macrolid polyene antibiotic that acts on mitochondria by inhibiting respiration and uncoupling. It also functions as an ionophore of Mg2+ in isolated rat liver mitochondria. I don’t find any information regarding olifen and mitochondrial function. Please check.
Line 932: Incomplete sentence “Activates the synthesis of endogenous antioxidant – glutathione”
Line 949: KoQ, not defined abbreviation.
Minor problems: Please check the use of abbreviations. For example ROS is defined at least 5 times throughout this manuscript.
Author Response
We thank the reviewer for the evaluation of our manuscript and constructive feedback.
We have made the revisions according to the suggestions. We have now corrected the title of the manuscript. In addition:
1. The concept of “mitochondrial dysfunction” is rather new, especially in cerebral ischemia. Different researchers have their own view on the mechanisms of mitochondrial dysfunction formation. The informativeness of molecular and biochemical markers has not been completely determined yet. In many works as biochemical markers of mitochondrial dysfunction are used indicators of energy metabolism. Therefore, this topic is inexhaustible in the coming decades.
2. We endeavored to adhere closely to our narrative plan, detailing the mechanisms of mitochondrial dysfunction, its role in neuroapoptosis, oxidative stress, and more. We may have deviated from the main storyline at times, but only to include supplementary, indirect information when the primary content was insufficient.
3. We have now added references to sources confirming the potential value of pyruvate dehydrogenase and malate dehydrogenase as markers of mitochondrial dysfunction.
4. We decided to retain menadione. Indeed, it exhibits both pro- and antioxidant properties and can protect or damage mitochondria. It is essential to familiarize specialists with the full profile of potential agents affecting mitochondrial dysfunction. Moreover, chemical modification of the menadione molecule may enhance its pharmacological value.
5. Oliphen or Hypoxen is an antihypoxant developed back in the 1970s. We have included information about it in the summary table of pharmacological agents affecting mitochondrial dysfunction.
6. The concept of neuroapoptosis is widely used in scientific works. For example, there is even a definition of "neuroapoptosis" in the glossary (Encyclopedia on Early Childhood Development, Glossary-Brain, February 4, 2011, ©2011 Centre of Excellence for Early Childhood Development).
Reviewer 3 Report
The authors present an insightful review on the molecular mechanism and potentially targets intervention to mitochondria but the main issue is occasional awkward English presentation and that can be overcome by English editing. Another issue that can be clarified is whether the authors are focusing on their discussion in CNS.
I am not entirely with the authors on their Introduction. how about citing some statistics to support the penetrance of mitochondrial abnormalities thus strengthening the importance of this topic?
It is not clear on the message illustrated in Fig1 - I guess the authors want to bring up the concept of primary vs secondary mitochondrial dysfunction leading to different pathology. The authors can make use of the figure legends to elaborate their points. is Fig2 not presenting the different pathways rather than the different disease types?
_
Author Response
We thank the reviewer for the evaluation of our manuscript and constructive feedback.
The authors have taken all the reviewer's suggestions into account and made corrections to the manuscript. In particular, the text was revised and proofread to improve the language quality. Additionally, we provided explanations for each figure, which will indeed enhance the significance of the work.
The inclusion of statistical data on the penetrance of primary mitochondrial dysfunction is indeed important and will be implemented in our next study on identifying primary mitochondrial dysfunction in newborns.
Round 2
Reviewer 1 Report
Having some blood tests for detecting mitochondrial dysfunction in stroke patients will be ideal. It looks like these need to be further investigated and are currently unavailable.
No more comments
Author Response
We thank the reviewer for this comment and that the reviewer agrees that this will be an interesting aspect of future studies.
Reviewer 2 Report
Please see detailed comments below.
Line 365-367: The presented model does not represent the current model of the permeability transition pore nor does it reflect the current discussion of the nature of this pore.
Line 964: This is an incomplete sentence: "Activates the synthesis of endogenous antioxidant - glutathione"
KoQ: undefined abbreviation
Reactive oxygen species (ROS): this abbreviation has been introduced a total of 15 times!!! One time is enough, please use the abbreviation after defining it.
Author Response
We thank the reviewer for additional feedback.
To date, the molecular structure of the mitochondrial pore has not been unequivocally identified. None of the proteins known to be part of the mitochondrial pore have proven to be absolutely essential for its function. Therefore, it is possible that different types of mitochondrial pores exist, or that more in-depth studies of the mitochondrial pore are needed. There are also three main models of the mitochondrial pore. Some believe that the mitochondrial pore lacks a distinct molecular basis, suggesting that several different pores may be composed of various components.
We can stylistically refine the definition of the mitochondrial pore as follows:
"The mitochondrial pore is a supramolecular channel connecting the cytosolic and intramitochondrial spaces, composed of a complex of proteins including the adenine nucleotide translocator, the benzodiazepine receptor (translocator protein), and the voltage-dependent anion channel. Cyclophilin D, ATPase, and the mitochondrial inorganic phosphate carrier are not part of the pore structure but act as regulatory factors."
We also revised the abbreviations accordingly